# CD38 promotes hematopoietic stem cell dormancy

**Liliia Ibneeva**[ID][1], **Sumeet Pal Singh**[2], **Anupam Sinha**[1], **Sema Elif Eski**[2], **Rebekka Wehner**[3,4,5], **Luise Rupp**[3], **Iryna Kovtun**[1], **Juan Alberto Pérez-Valencia**[1], **Alexander Gerbaulet**[3], **Susanne Reinhardt**[6], **Manja Wobus**[5,7], **Malte von Bonin**[7], **Jaime Sancho**[8], **Frances Lund**[9], **Andreas Dahl**[6], **Marc Schmitz**[3,4,5], **Martin Bornhäuser**[4,5,7], **Triantafyllos Chavakis**[1], **Ben Wielockx**[1,10‡], **Tatyana Grinenko**[ID][1,11‡]*

1 Institute for Clinical Chemistry and Laboratory Medicine, University Hospital and Faculty of Medicine, Technische Universität Dresden, Dresden, Germany, 2 IRIBHM, Université Libre de Bruxelles (ULB), Brussels, Belgium, 3 Institute for Immunology, Faculty of Medicine Carl Gustav Carus, Technische Universität Dresden, Dresden, Germany, 4 National Center for Tumor Diseases (NCT), Partner Site Dresden, Dresden, Germany, 5 German Cancer Consortium (DKTK), Partner Site Dresden, and German Cancer Research Center (DKFZ), Heidelberg, Germany, 6 DRESDEN-concept Genome Center, Center for Molecular and Cellular Bioengineering, Technische Universität Dresden, Dresden, Germany, 7 Medical Clinic I, University Hospital Carl Gustav Carus, Technische Universität Dresden, Dresden, Germany, 8 Instituto de Parasitología y Biomedicina "López-Neyra" CSIC, Granada, Spain, 9 Department of Microbiology, University of Alabama at Birmingham, Birmingham, Alabama, United States of America, 10 Experimental Center, Faculty of Medicine, Technische Universität Dresden, Dresden, Germany, 11 Shanghai Institute of Hematology, State Key Laboratory of Medical Genomics, National Research Center for Translational Medicine at Shanghai, Ruijin Hospital Affiliated to Jiao Tong University School of Medicine, Shanghai, China

‡ These authors are joint senior authors on this work.
* Tatyana.grinenko@uniklinikum-dresden.de

**Data Availability Statement:** Bulk and single-cell RNA-sequencing data are available at GEO under accession numbers GSE196760 and GSE196759, respectively. All data related to the Figures can be found in S1 Data.

## Abstract

A subpopulation of deeply quiescent, so-called dormant hematopoietic stem cells (dHSCs) resides at the top of the hematopoietic hierarchy and serves as a reserve pool for HSCs. The state of dormancy protects the HSC pool from exhaustion throughout life; however, excessive dormancy may prevent an efficient response to hematological stresses. Despite the significance of dHSCs, the mechanisms maintaining their dormancy remain elusive. Here, we identify CD38 as a novel and broadly applicable surface marker for the enrichment of murine dHSCs. We demonstrate that cyclic adenosine diphosphate ribose (cADPR), the product of CD38 cyclase activity, regulates the expression of the transcription factor c-Fos by increasing the release of $Ca^{2+}$ from the endoplasmic reticulum (ER). Subsequently, we uncover that c-Fos induces the expression of the cell cycle inhibitor $p57^{Kip2}$ to drive HSC dormancy. Moreover, we found that CD38 ecto-enzymatic activity at the neighboring CD38-positive cells can promote human HSC quiescence. Together, CD38/cADPR/$Ca^{2+}$/c-Fos/$p57^{Kip2}$ axis maintains HSC dormancy. Pharmacological manipulations of this pathway can provide new strategies to improve the success of stem cell transplantation and blood regeneration after injury or disease.

**Funding:** This work was supported by a grant from the Deutsche Forschungsgemeinschaft (GR 4857/2-1) to T.G. and Deutsche Forschungsgemeinschaft (WI3291/5-1, 12-1 and 13-1) to B.W. T.C. is supported by the Deutsche Forschungsgemeinschaft (TRR332, project B4). S.P.S. was supported by Fonds National de la Recherche Scientifique (40005588 – MISU-PROL) and Jaumotte-Demoulin Foundation, S.E.E. was supported by PhD Fellowship from Fonds National de la Recherche Scientifique (40006730 – ASP). T.G. was supported by Mildred-Scheel-Nachwuchszentrum fellowship. The funders had no role in the study design, data collection and analysis, and preparation of the manuscript.

**Competing interests:** The authors have declared that no competing interests exist.

**Abbreviations:** ADPR, adenosine diphosphate ribose; cADPR, cyclic adenosine diphosphate ribose; CMP, common myeloid progenitor; dHSC, dormant hematopoietic stem cell; ER, endoplasmic reticulum; FBS, fetal bovine serum; GMP, granulocyte-monocyte progenitor; GSEA, gene set enrichment analysis; hHSC, human hematopoietic stem cell; HSC, hematopoietic stem cell; LT-HSC, long-term hematopoietic stem cell; MEP, megakaryocyte-erythroid progenitor; MNC, mononuclear cell; MMP, mitochondrial membrane potential; NAD, nicotinamide adenine dinucleotide; PB, peripheral blood; PBS, phosphate-buffered saline; TBM, total bone marrow; TGF-β1, transforming growth factor beta 1; TNFα, tumor necrosis factor alpha.

## Introduction

Hematopoietic stem cells (HSCs) are responsible for the production of all blood cells during life. Most adult HSCs are maintained in a quiescent state; furthermore, numerous studies have demonstrated that 20% to 30% of murine HSCs are deeply quiescent and, therefore, called "dormant" HSCs (dHSCs) [1]. dHSC are characterized by slow metabolism, reduced ribosomal biogenesis, and DNA replication [1]. Mechanisms, which keep dHSC in a quiescent state, protect them not only from external stresses [2,3] but also from the accumulation of somatic mutations to prevent their exhaustion or malignant transformation [1,4–6]. Despite their dormancy, dHSCs harbor the greatest long-term repopulation capacity in transplantation assays [1,7,8]. They do not produce cells under homeostatic conditions and are activated only in response to severe stress signals such as interferons, lipopolysaccharide, and myeloablation [1,9]. Thus, dHSCs serve as a reserve pool of stem cells throughout life.

However, despite the importance of dormant HSCs, their detailed characterization has been challenging due to the absence of known surface markers for their ready identification and isolation. Consequently, processes involved in the preservation of dHSC quiescence are poorly understood. Recently, Cabezas-Wallscheid and colleagues have established a Gprc5c (retinoic acid-induced gene) reporter mouse strain and have shown that retinoic acid signaling and hyaluronic acid could regulate HSC dormancy [8,10]. Fukushima and colleagues used another p27 (Cdk inhibitor) reporter mouse strain to reveal that HSC entry in to the cell cycle is controlled by Cdk4/6 and that high cytosolic $Ca^{2+}$ concentration correlates with HSC quiescence [11]. However, why dHSCs harbor high cytosolic $Ca^{2+}$ concentration, and how $Ca^{2+}$ regulates HSC dormancy remain unclear. In the present study, we identify CD38 as a surface marker for the isolation of murine dHSCs and describe a previously unknown signaling axis driven by the ecto-enzymatic activity of CD38 controlling HSC dormancy. Mechanistically, we show that cyclic adenosine diphosphate ribose (cADPR), the product of nicotinamide adenine dinucleotide (NAD) conversion by CD38, regulates the expression of the transcription factor c-Fos, thereby driving quiescence in a p57$^{Kip2}$-dependent manner. Moreover, we found that CD38 ecto-enzymatic activity at the neighboring CD38-positive cells promotes human HSC quiescence, which are CD38 negative.

Results

### Pseudotime analysis of HSCs reveals the transition between proliferation and dormancy

To capture the transition between quiescence and proliferation, HSCs from young mice were subjected to single-cell RNA sequencing, and, after quality control, the transcriptome profiles of 1,613 individual HSCs were used for downstream analysis (Figs 1A, S1A, and S2A). To identify actively cycling cells, we calculated cell cycle and dormancy scores of individual HSCs using Seurat [12], which were based on the expression levels of cell cycle and dormancy genes [13] (S1 Table). We observed that cells in the S (Fig 1B) and the G2/M phases (Fig 1C) were clustered together and that, as expected, most of the HSCs were quiescent (Fig 1D). Further, comparison of Fig 1A with Fig 1B–1D showed that pseudotime ordering was congruent with the transition from proliferation to dormancy. Next, we applied pseudotime ordering (Fig 1A) to identify gene expression patterns that correlate with cell cycle dynamics (Fig 1E) and identified 3 major gene expression clusters, namely, 1—Early, 2—Intermediate, and 3—Late genes, according to peak expression in relation to pseudotime (Fig 1E and S2 Table). Functional annotation of each cluster revealed that Intermediate and Late genes were typically related to cell cycle activation pathways (Fig 1F and S3 Table). In contrast, Early genes included well-known markers of HSCs with high transplantation potential, i.e., *Vwf* [14], *Procr* [15], *Fgd5*

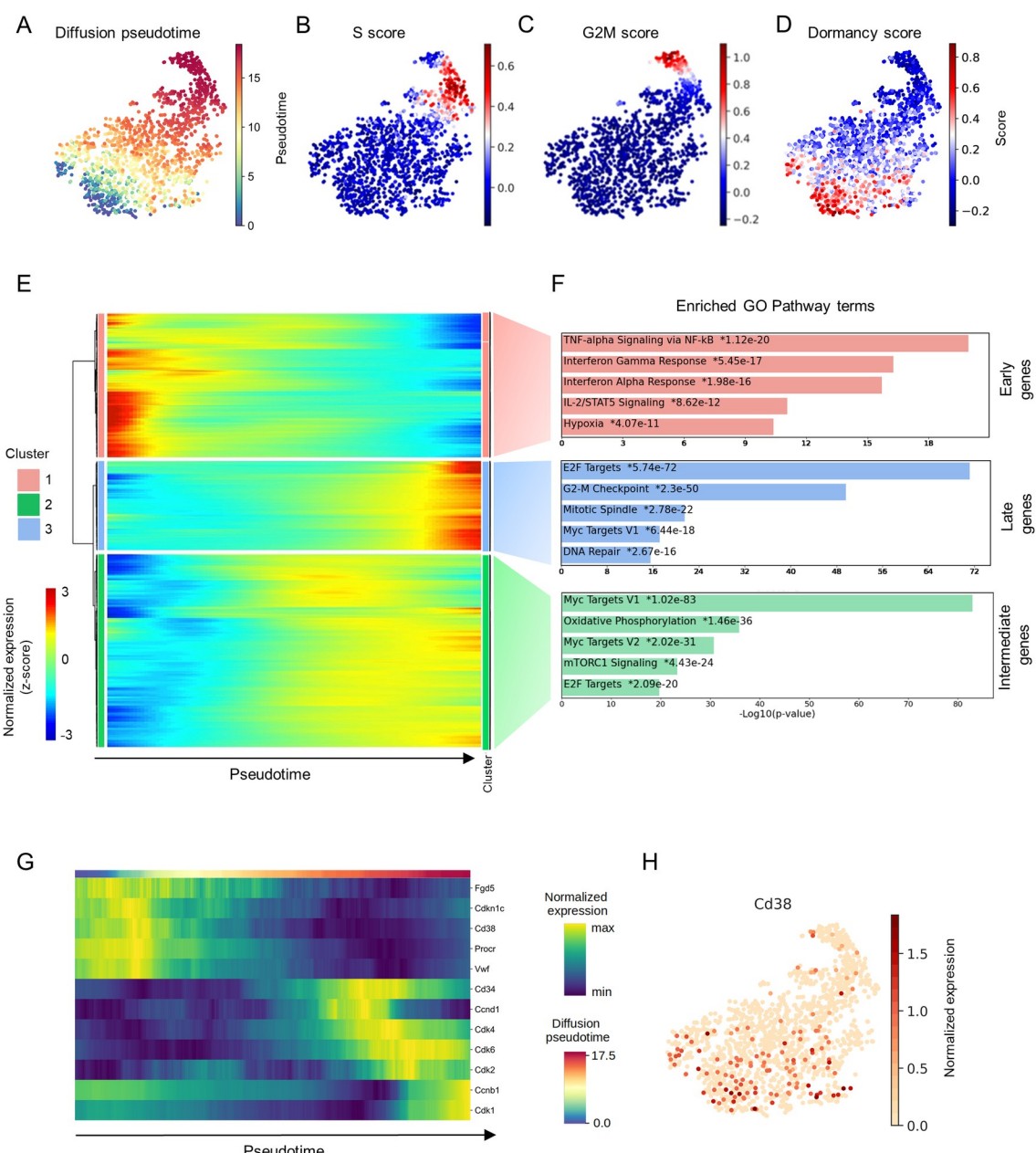

**Fig 1. Single-cell transcriptome analysis of HSCs.** (A). UMAP representation depicting the transcriptional profiles of individual HSCs (LSK CD48⁻ CD150⁺). (B) S-phase score along pseudotime. (C) G2/M phase score along pseudotime. (D) Dormancy score along pseudotime. For panels A–D, each dot represents a single cell. (E) Clustered heatmap showing expression of genes along pseudotime. Each column represents a single cell and each row represents a single gene. (F) The first 5 most significantly enriched pathways in each cluster are shown. (G) Expression of selected genes along pseudotime. (H) UMAP representation showing the expression of *Cd38*. Each dot represents a single cell. The data underlying this figure can be found in S1–S3 Tables. HSC, hematopoietic stem cell; UMAP, uniform manifold approximation projection.

[16,17], and the cell cycle inhibitor *Cdkn1c* [18,19] (Figs 1G, S1B, and S1C). Notably, this cluster was characterized by genes involved in pathways related to the activation of tumor necrosis factor alpha (TNFα) signaling, interferon gamma and alpha response, Stat3 and Stat5, as well as transforming growth factor beta 1 (TGF-β1) signaling, which is a well-known regulator of HCS quiescence [20] (Fig 1F and S3 Table). Next, we attempted to isolate cell surface markers

within the group of Late genes (S2 Table) associated with HSC dormancy and identified *Cd38* as a putative marker for dHSCs because its expression was higher in cells with high dormancy scores and corresponded with expression of well-known long-term HSC (LT-HSC) markers (Fig 1G and 1H).

## CD38$^+$ LT-HSCs harbor the highest repopulation capacity

We analyzed surface expression of CD38 on hematopoietic stem and progenitor cells (HSPCs) and showed that CD38 was expressed by fractions of LT-HSCs (Lin$^-$c-Kit$^+$ Sca-1$^+$ (LSK) CD48$^-$ CD150$^+$ CD34$^-$ CD201$^+$; 36.6 ± 2.5%), HSCs (LSK CD48$^-$ CD150$^+$; 12.4 ± 0.7%), and multipotent progenitors 2 (MPP2, LSK CD48$^+$ CD150$^+$; 15.3 ± 1.8%) but not short-term HSCs (ST-HSCs, LSK CD48$^-$ CD150$^-$) or multipotent progenitors 3/4 (MPP3/4, LSK CD48$^+$ CD150$^-$) (Figs 2A–2B and S2A–S2B). We demonstrated that total bone marrow (TBM) cells can be used to define the CD38$^+$ fraction in the absence of CD38 knock-out mice (CD38KO) (S2C Fig), providing the possibility of an internal positive control for easy identification of CD38$^+$ cells. Next, we subdivided HSCs based on CD38 surface expression as CD38$^+$ and CD38$^-$ stem cells and compared the expression of well-known surface markers defining the most potent LT-HSCs [1,21–25], and revealed that, compared to CD38$^-$ HSCs, surface expression of CD34, CD229, and c-Kit were lower, while that of CD201, Sca-1, and CD150 were higher in CD38$^+$ HSCs (S2D Fig). In agreement with these data, the frequency of LT-HSCs was higher among CD38$^+$ HSCs compared to other fractions of the HSCs (Fig 2C). Taken together, these data indicate that CD38$^+$ HSCs display a phenotype of the most potent and quiescent LT-HSCs [1,21–25].

Only a fraction of LT-HSCs ((LSK) CD48$^-$ CD150$^+$ CD34$^-$ CD201$^+$) expresses CD38. To assess whether CD38 expression correlates with the superior repopulation capacity within LT-HSCs, we transplanted CD38$^+$ and CD38$^-$ LT-HSCs into lethally irradiated mice under competitive settings (Figs 2D and S3A). While CD38$^-$ LT-HSCs produced more short-lived neutrophils 4 weeks after transplantation, CD38$^+$ LT-HSCs repopulated the HSC compartment and peripheral blood (PB) more efficiently at 20 weeks after primary transplantation and after secondary transplantation as well (Fig 2E–2G). Further, no lineage bias was observed in the reconstitution pattern of CD38$^+$ and CD38$^-$ LT-HSCs (Figs 2H and S3B). These results demonstrate the superior repopulation and self-renewal capacity of CD38$^+$ cells compared to CD38$^-$ LT-HSCs.

To understand the hierarchy between CD38$^+$ and CD38$^-$ LT-HSCs, we compared the expression of CD38 on the progeny of donor HSCs and found that, while CD38$^+$ LT-HSCs gave rise to both CD38$^-$ and CD38$^+$ HSCs, CD38$^-$ cells could not generate CD38$^+$ HSCs (Figs 2I, S3C and S3D). Taken together, these results indicate that CD38$^+$ LT-HSCs reside at the top of the hematopoietic cell hierarchy. We propose that the CD38 surface marker can be used atop to the well-established immuno-phenotype of LSK CD48$^-$ CD150$^+$ CD34$^-$ CD201$^+$ to define the most potent LT-HSCs.

## High levels of surface CD38 expression define the LT-HSCs population enriched with dHSCs

We performed cell cycle analyses, bromodeoxyuridine (BrdU) incorporation, and long-term label-retaining assays to investigate whether CD38 expression correlates with stem cells' dormancy (Fig 3A–3E). While LT-HSC markers already enrich for more quiescent cells, CD38$^+$ LT-HSCs contained an even higher frequency of cells in G0 phase than CD38$^-$ LT-HSCs (Fig 3A). Accordingly, CD38$^+$ LT-HSCs incorporated BrdU significantly slower and retained higher levels of H2B-GFP after 130 days of chase compared with CD38$^-$ cells (Fig 3B–3E),

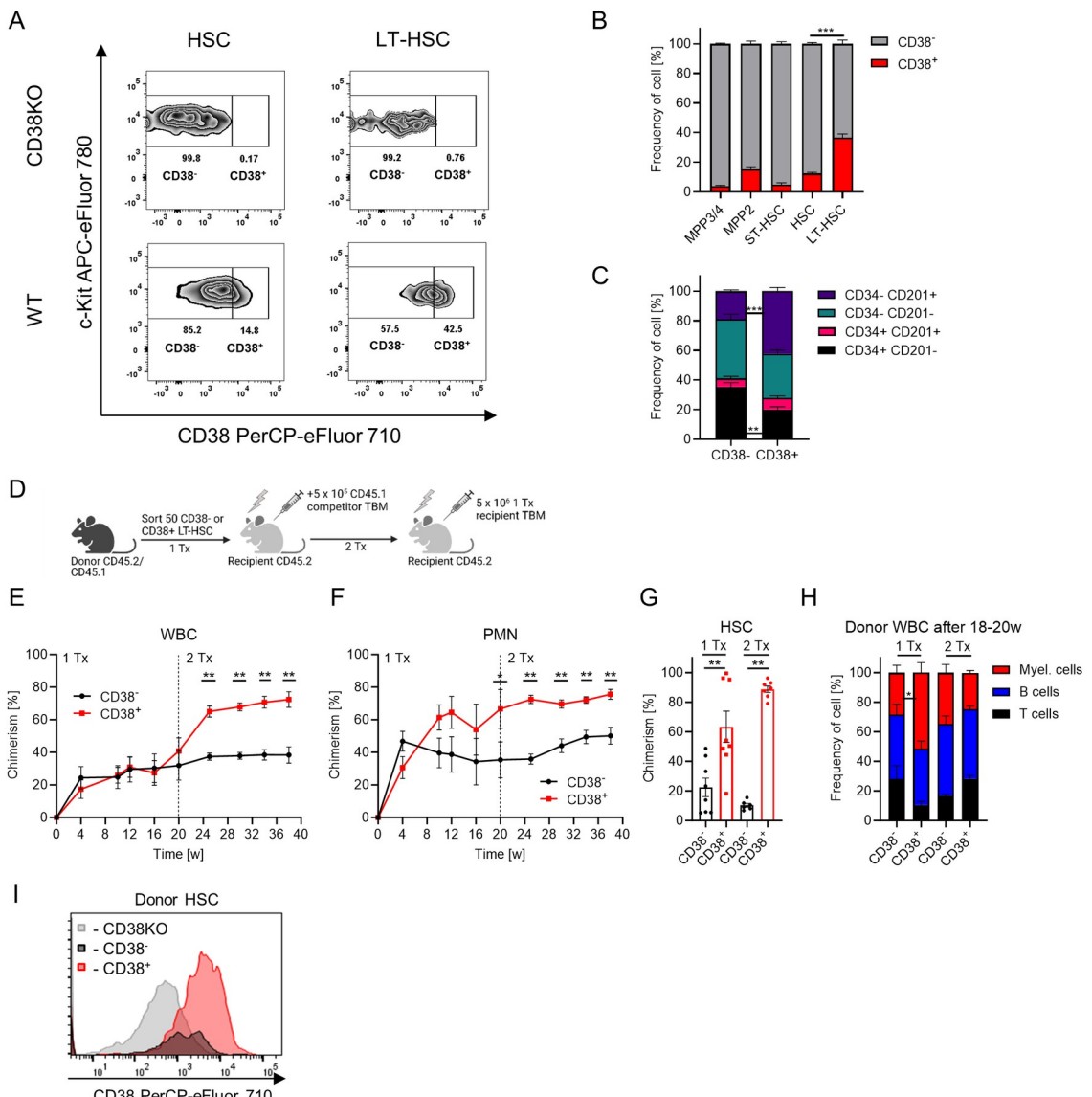

**Fig 2. CD38⁺ defines LT-HSCs with the highest long-term repopulation capacity.** (A) Flow cytometry analysis of CD38 expression on HSCs (LSK CD48⁻ CD150⁺) and LT-HSCs (LSK CD48⁻ CD150⁺ CD34⁻ CD201⁺) in CD38KO (negative control for staining) vs. wt mice. (B) Frequency of CD38⁺ cells in various HSPC populations, $n = 7$. Multiple-group comparisons were performed using Brown–Forsythe and Welch ANOVA followed by Dunnett's T3 multiple comparison tests. ***$p < 0.001$. (C) Frequencies of different HSC subpopulations in CD38⁻ and CD38⁺ HSCs, $n = 7$. (D) Set-up for CD38⁻ and CD38⁺ LT-HSCs (LSK CD48⁻ CD150⁺ CD201⁺ CD34⁻) transplantation, 2 independent experiments, 1Tx—$n = 8$, 2Tx—$n = 6$ vs. 7. Created with BioRender.com. (E) Chimerism in donor-derived WBC cells after transplantation. (F) Chimerism in donor-derived PMNs after transplantation, PMN: Gr1⁺ CD11b⁺. (G) Chimerism in the HSC population after transplantation. (H) Frequency of T, B, and myeloid cells in donor-derived PB cells at 18–20 weeks after transplantation. (I) Surface expression of CD38 in donor-derived HSCs at 20 weeks after primary transplantation of CD38⁺ or CD38⁻ LT-HSCs (the same amount of cells from each group was pooled together and stained in a single tube using the same antibody master mix, 1 representative experiment is shown, see data from the second experiment in S3C Fig), same amount of CD38 knock-out HSCs was used as negative control for CD38 staining. C, E–H—$p$-value was calculated using Mann–Whitney $U$-test, *$p < 0.05$, **$p < 0.01$. The data underlying this figure can be found in S1 Data. LT-HSC, long-term hematopoietic stem cell; PB, peripheral blood; PMN, polymorphonuclear neutrophil.

revealing that CD38⁺ LT-HSCs are more deeply in state of quiescence than their CD38⁻ counterparts [26]. Moreover, CD38⁺ LT-HSCs had lower mitochondrial membrane potential (MMP) than CD38⁻ stem cells, despite no difference in mitochondrial mass (S3E and S3F Fig),

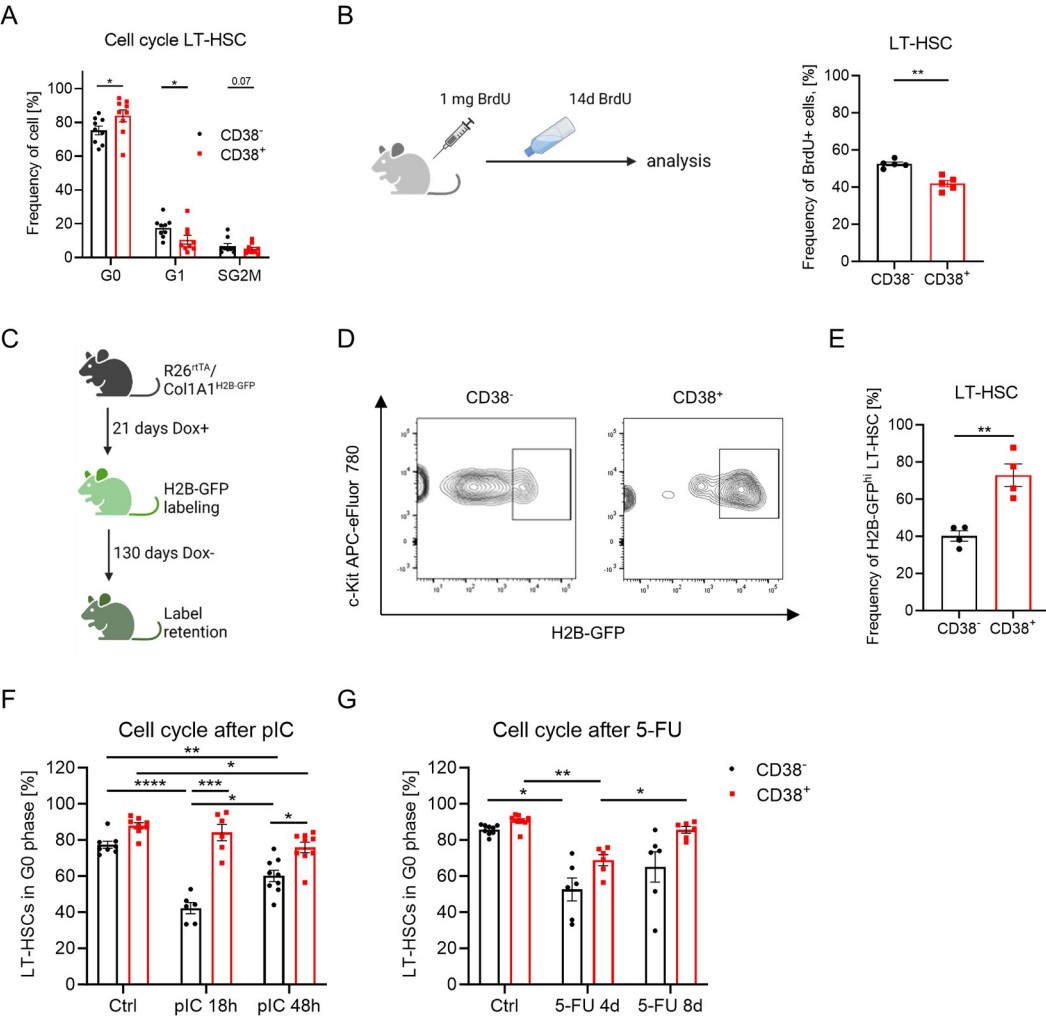

**Fig 3. CD38 regulates LT-HSC quiescence.** (A) Frequencies of CD38$^-$ and CD38$^+$ LT-HSCs in G0, G1, and SG2M phases of the cell cycle, $n = 9$. (B) BrdU incorporation assay. Frequency of BrdU$^+$ cells in CD38$^-$ and CD38$^+$ LT-HSC populations 14d after BrdU, $n = 5$. (C) Set-up of a long-term label retention assay. (D) Representative plot defining H2B-GFP$^+$ cells in LT-HSCs. (E) Frequency of GFP$^+$ cells in CD38$^-$ and CD38$^+$ LT-HSCs, $n = 4$. (F) Cell cycle analysis of LT-HSCs 18 h and 48 h after pIC injection ($n = 6$–9, 3 independent experiments). (G) Cell cycle analysis of LT-HSCs 4d and 8d after 5-FU injection ($n = 6$–9, 3 independent experiments). For panels A, B, E, Mann–Whitney $U$-test was used. Multiple-group comparisons were performed using Brown–Forsythe and Welch ANOVA followed by Dunnett's T3 multiple comparison tests. $^*p < 0.05$, $^{**}p < 0.01$, $^{***}p < 0.001$, $^{****}p < 0.0001$. The data underlying this figure can be found in S1 Data. Graphical schemes in panels B and C were created with BioRender.com. LT-HSC, long-term hematopoietic stem cell.

which is in agreement with previous findings that HSC quiescence is associated with a lower metabolic status [8,27].

To investigate CD38$^+$ LT-HSCs' cell cycle activation in response to hematopoietic stresses, we treated mice with polyI: polyC (pIC) mimicking viral infection [8], and the myeloablative agent 5-fluorouracil (5-FU). CD38$^-$ LT-HSC rapidly entered the cell cycle in response to pIC, while CD38$^+$ LT-HSCs started cycling only 48 h after pIC injection (Fig 3F). In the 5-FU chemotherapy model, both CD38$^-$ and CD38$^+$ LT-HSCs actively proliferated 4 days after 5-FU injection (Fig 3G). Although CD38$^+$ LT-HSCs tended to restore their quiescence 8 days after 5-FU injection, CD38$^-$ stem cells remained cycling. Therefore, CD38$^+$ LT-HSCs required more time to enter the cell cycle in response to hematological stresses while returning to the

quiescent state faster than CD38⁻ cells. Taken together, we conclude that CD38 can be used for the enrichment of dHSCs [1,8].

## CD38 enzymatic activity regulates HSC quiescence

To understand whether CD38 is directly involved in the maintenance of dHSCs, we compared long-term repopulation and self-renewal capacities of wild-type (wt) and CD38KO LT-HSCs. We did not find any difference in composition of PB or bone marrow at steady state (S4A–S4F Fig). Although only about 40% of LT-HSC in wt mice express CD38, we found that long-term repopulation and self-renewal capacity of CD38KO LT-HSC were lower than those of wt cells (Fig 4A–4E). In agreement with this finding, CD38KO TBM cells had diminished long-term repopulation capacity compared with wt TBM (Fig 4F–4J). While the analysis of LT-HSC populations did not reveal any cell cycle differences between wt and CD38KO (Fig 4K), CD38⁺ LT-HSCs were more quiescent than CD38KO counterparts (Fig 4L). To investigate the possible LT-HSC cell cycle regulation by BM niche, we transplanted CD38KO cells into wt and CD38KO recipients. However, CD38 from the BM microenvironment did not influence the cell cycle of CD38KO HSCs (S4G Fig). Taken together, these results suggest that CD38 supports the functionality of LT-HSCs.

CD38 is a multifaceted NAD catabolic ecto-enzyme that metabolizes NAD and its precursors (nicotinamide mononucleotide-NMN and nicotinamide riboside-NR) into adenosine diphosphate ribose (ADPR) and cyclic-ADPR (cADPR) [28]; 78c is a specific CD38 inhibitor that hinders both hydrolase and ADP-ribosyl cyclase activities of CD38 [29]. To investigate whether the enzymatic activity of CD38 regulates the quiescence of LT-HSCs, we performed single-cell tracing experiment wherein LT-HSC division in the presence of 78c was tracked (Figs 5A–5C and S5A–S5C). In agreement with our previous data (Fig 3A–3E), CD38⁺ cells were more quiescent than CD38⁻ LT-HSCs, whereas inhibition of CD38 by 78c accelerated the first division of CD38⁺ but not CD38⁻ LT-HSCs or LT-HSCs from CD38KO mice (Figs 5A–5C and S5A–S5C). In line with this finding, treatment with 78c activated the cell cycle entrance of CD38⁺ HSCs in vitro (S5D Fig) and delayed the restoration of CD38⁺ LT-HSC quiescence after 5-FU treatment in vivo (Fig 5D). Together, our data support the idea that CD38 enzymatic activity contributes to the maintenance of LT-HSC quiescence.

Human HSCs (hHSCs) are defined as CD38$^{lo/-}$ cells [30]. Additionally, we found that low surface expression of CD38 correlates with low intracellular CD38 levels in Lin⁻ CD34⁺ CD38$^{lo/-}$ hHSCs (S6A and S6B Fig). As expected, the inhibition of CD38 enzymatic activity did not influence CD38$^{lo/-}$ hHSC cell cycle activation (S8C–S8E Fig). We hypothesized that the CD38 ecto-enzymatic activity at the neighboring CD38-positive cells may regulate hHSC quiescence. We cultured hHSCs (Lin⁻ CD34⁺ CD38$^{lo/-}$) together with CD38⁺ tumor cell line or CD38⁺ progenitors (Figs 5E and S6E) and found that inhibition of CD38 enzymatic activity by 78c inhibitor (S6D Fig) led to the cell cycle entrance of hHSCs (Figs 5E and S6E). Moreover, we analyzed healthy human bone marrow and showed that Ki67⁻ quiescent hHSCs localized significantly closer to the CD38⁺ cells than Ki67⁺ active hHSCs (Fig 5F). Therefore, we conclude that CD38 can regulate hHSC quiescence in a paracrine manner.

## cADPR supports high cytoplasmic [Ca²⁺] levels in CD38⁺ HSCs

Both products of CD38 enzymatic activity, i.e., ADPR and cADPR, increase cytosolic $Ca^{2+}$ concentration ($[Ca^{2+}]_c$) in several cell types [31–33] (Fig 6A) and high cytoplasmic $Ca^{2+}$ has been shown to support quiescence of HSCs [11]. Consistently, we show that $[Ca^{2+}]_c$ is indeed higher in CD38⁺ LT-HSCs than in CD38⁻ or CD38KO cells when measured using $Ca^{2+}$-indicator dyes, Fluo-8AM and Indo-1 (Fig 6B and 6C). Additionally, while treatment of cells with

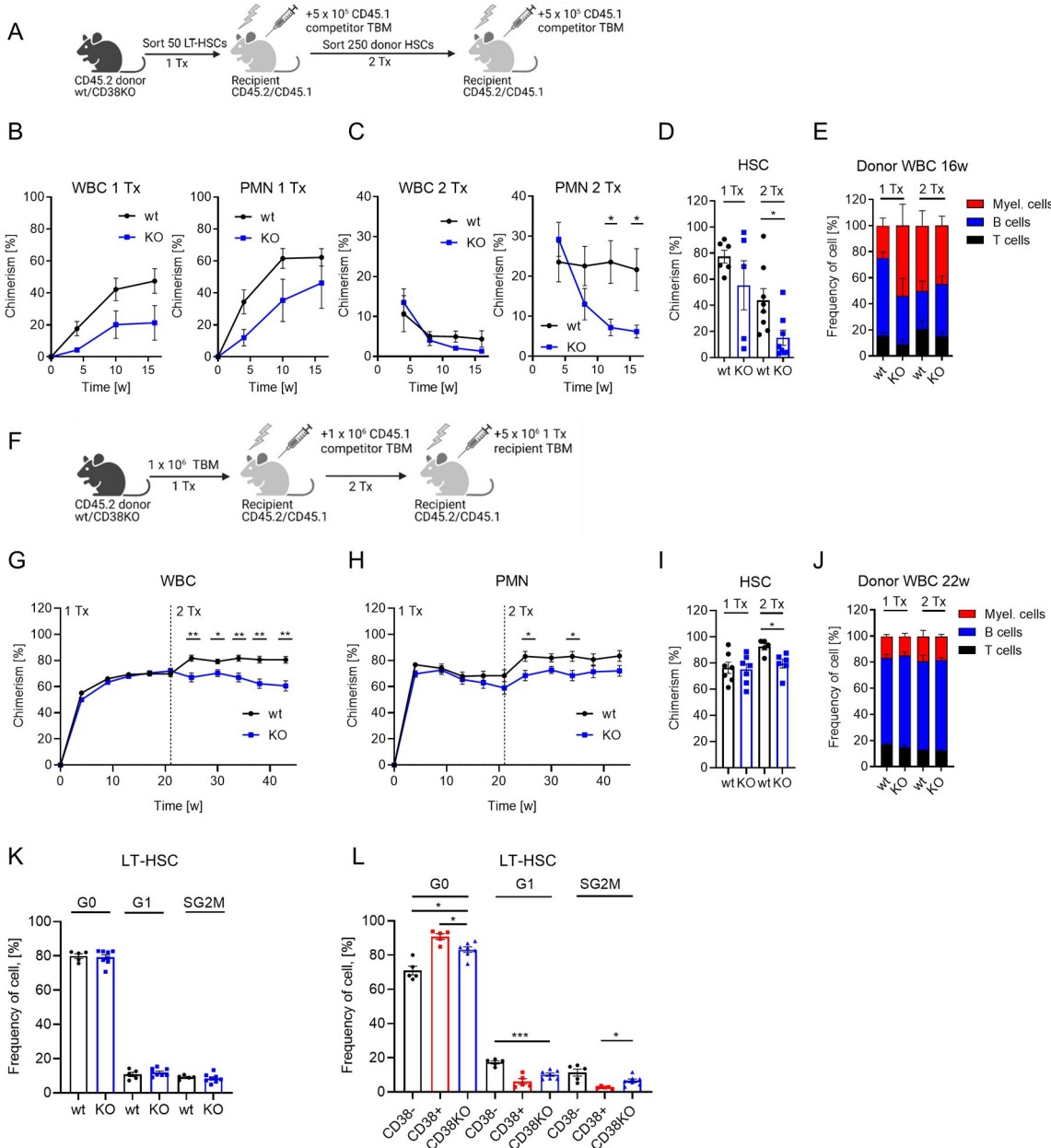

**Fig 4. Reduced repopulation capacity of LT-HSCs from CD38KO mice.** (A) Experimental setup for transplantation of LT-HSCs from wt and CD38KO mice, 2 independent experiments, 1 representative experiment is shown. (B) Chimerism in donor-derived WBC and PMN (Gr1+ CD11b+) in PB of recipients after primary transplantation ($n = 5$–6). (C) Chimerism in donor-derived WBC and PMN in PB of recipients after secondary transplantation ($n = 8$). (D) Chimerism in donor-derived HSCs in bone marrow 16 weeks after transplantation. (E) Frequency of myeloid, B, and T cells in donor-derived WBCs 16 weeks after transplantation. (F) Experimental setup for transplantation of TBM from wt and CD38KO mice (1Tx—$n = 7$, 2Tx—$n = 5$–6). (G) Chimerism in donor-derived WBC and (H) PMN (Gr1+ CD11b+) in PB of recipients after primary and secondary transplantation. (I) Chimerism in donor-derived HSCs in bone marrow after primary and secondary transplantation. (J) Frequency of myeloid, B, and T cells in donor-derived WBCs 21–22 weeks after transplantation. (K) Frequencies of LT-HSCs in G0, G1, and SG2M phases of the cell cycle (wt $n = 5$, CD38KO $n = 8$). (L) Frequencies of CD38+ and CD38- LT-HSCs from wt and LT-HSC from CD38KO in G0, G1, and SG2M phases of the cell cycle (wt $n = 5$, CD38KO $n = 7$). $P$-values were calculated using Mann–Whitney $U$-test, *$p < 0.05$, **$p < 0.01$, ***$p < 0.001$. The data underlying this figure can be found in S1 Data. Graphical schemes in panels A and F were created with BioRender.com. LT-HSC, long-term hematopoietic stem cell; PB, peripheral blood; TBM, total bone marrow.

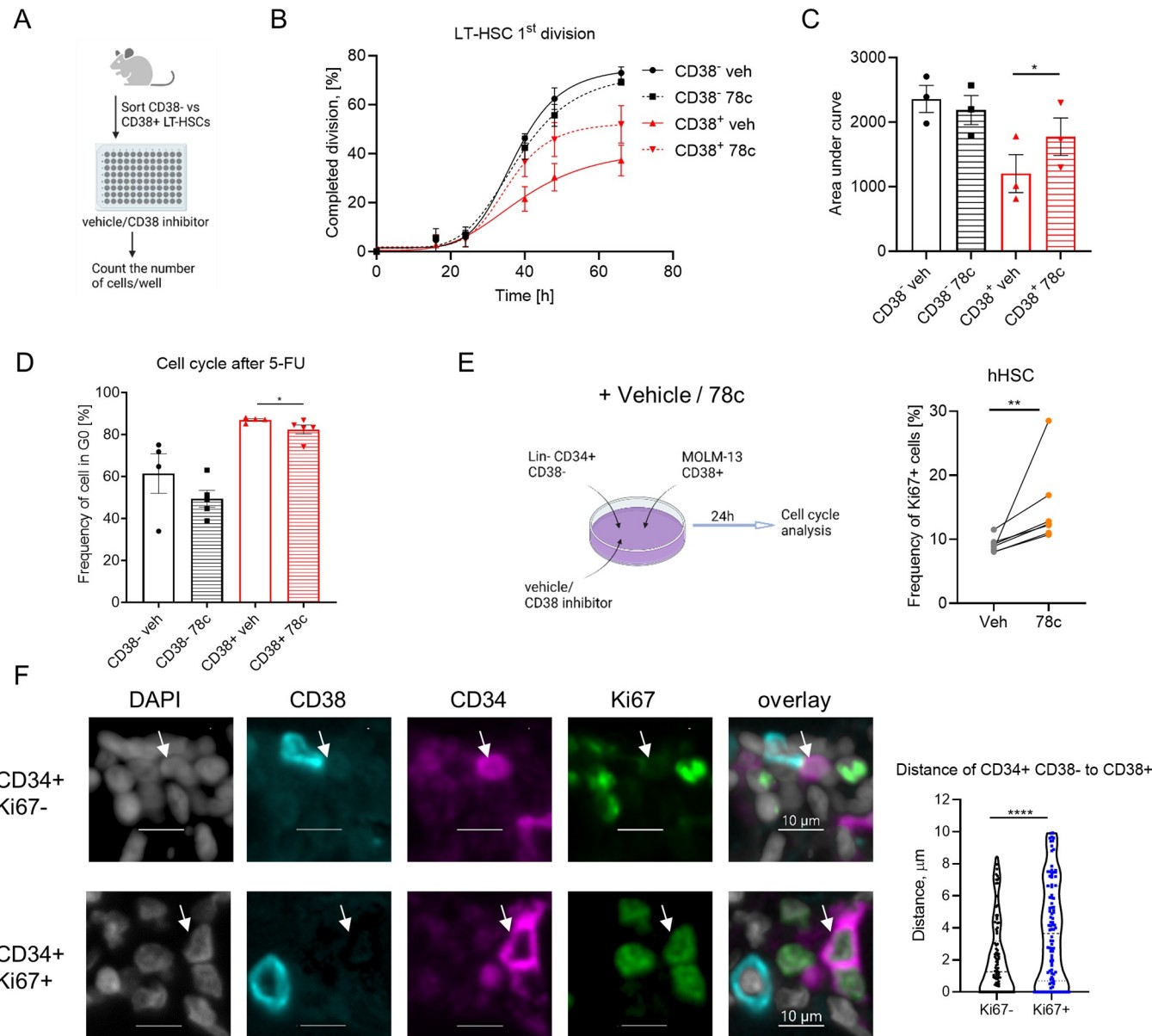

**Fig 5. CD38 enzymatic activity regulates LT-HSC exit from quiescence.** (A) Single CD38⁻ and CD38⁺ LT-HSCs were sorted and cultured in liquid media with or without 78c, a CD38 inhibitor. (B) Frequency of LT-HSCs that had completed the first division during incubation time is presented (3 independent experiments). (C) Quantification of the AUC for data in panel B. (D) Cell cycle analysis of CD38⁺ and CD38⁻LT-HSCs 8d after 5-FU and 78c injections. (E) Co-culture of hHSCs (Lin⁻ CD34⁺ CD38$^{lo/-}$) from healthy adult donors and MOLM-13 in the presence of CD38 inhibitor 78c for 24h. Cell cycle of hHSCs was analyzed ($n = 7$). (F) Representative picture of healthy human bone marrow multiplex immunofluorescence staining for CD34 (magenta), CD38 (cyan), Ki67 (green), and DAPI (gray). Scale bar indicates 10 μm. Distance between Ki67⁻ CD34⁺ CD38⁻ and Ki67⁺ CD34⁺ CD38⁻ to the closest CD38⁺ cell was analyzed ($n = 98$ Ki67⁻, 100 Ki67⁺). White arrow indicates CD34⁺ CD38⁻ cell. For panels C and E, the paired $t$ test was used. For panel D, the Mann–Whitney $U$-test was used. For panel F, the $P$-values were calculated using unpaired $t$ test. $^*p < 0.05$, $^{**}p < 0.01$, $^{****}p < 0.0001$. The data underlying this figure can be found in S1 Data. Graphical schemes in panels A and E were created with BioRender.com. AUC, area under the curve; hHSC, human hematopoietic stem cell; LT-HSC, long-term hematopoietic stem cell.

8-Bromo-ADPR (a cell-permeable antagonist that blocks ADPR-dependent Ca$^{2+}$ release) did not influence either [Ca$^{2+}$]$_c$ or cell cycle activity of HSCs (S7A and S7B Fig), blocking cADPR-dependent Ca$^{2+}$ release from the endoplasmic reticulum (ER) by the 8-Bromo-cADPR antagonist reduced [Ca$^{2+}$]$_c$ in HSCs (Fig 6D) and promoted their cell cycle entry (Fig 6E). Moreover,

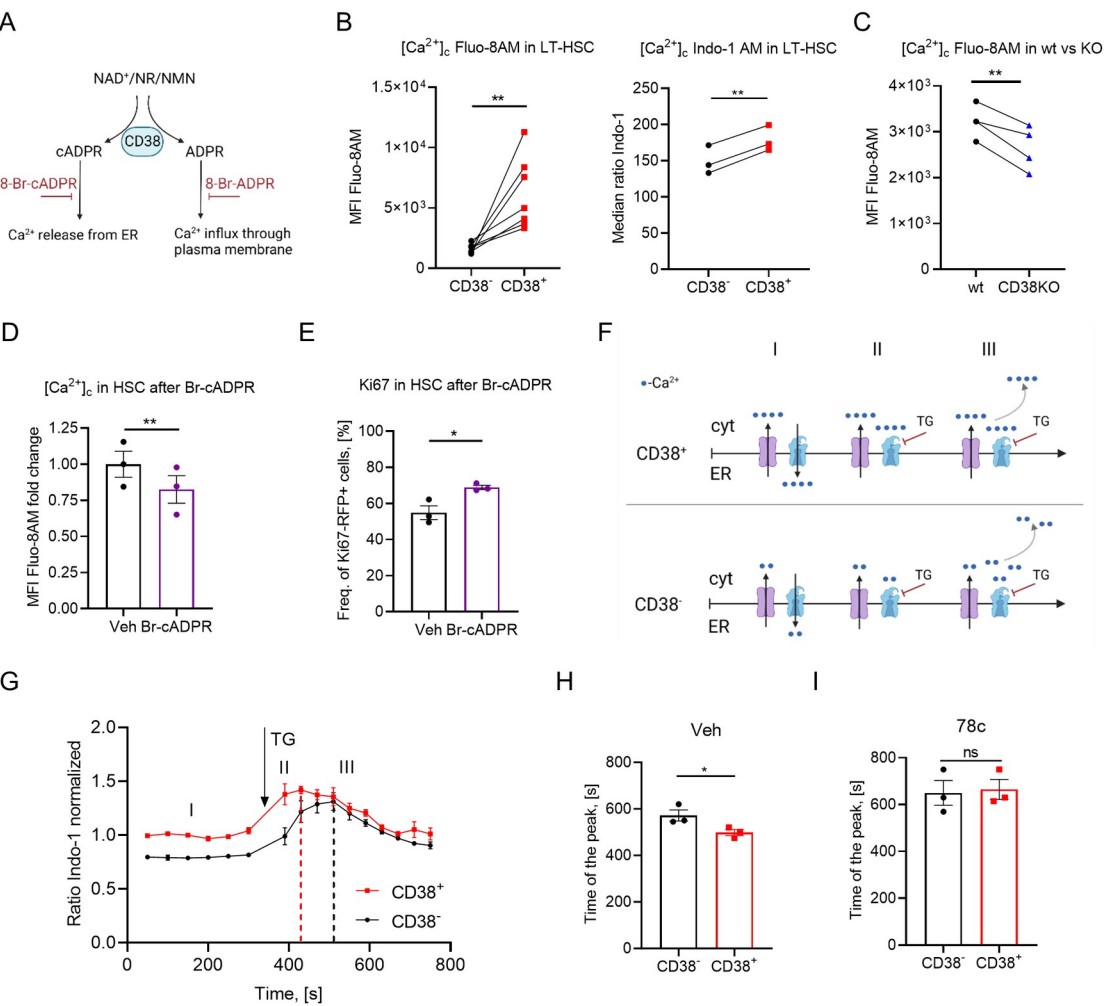

**Fig 6. CD38 enzymatic activity regulates cytoplasmic [Ca$^{2+}$].** (A) Schematic representation of enzymatic activities of CD38. (B) Free cytoplasmic Ca$^{2+}$ ([Ca$^{2+}$]$_c$) relative concentration in CD38$^+$ and CD38$^-$ LT-HSCs analyzed using Fluo-8AM dye, $n = 7$ (left), and ratiometric Indo-1 dye, $n = 3$ (right). (C) Free cytoplasmic Ca$^{2+}$ ([Ca$^{2+}$]$_c$) relative concentration in CD38$^+$ wt and LT-HSCs from CD38KO analyzed using Fluo-8AM dye, $n = 4$. (D) $5 \times 10^4$ LSK from Ki67$^{RFP}$ reporter mice were sorted and cultured for 24 h in the presence of cADPR antagonist (Br-cADPR), 2 independent experiments. Relative [Ca$^{2+}$]$_c$ concentration in HSCs treated with Br-cADPR for 24 h, $n = 3$. Two independent experiments and (E) frequency of cycling Ki67-RFP$^+$ HSCs at 24 h after treatment with Br-cADPR, $n = 3$, 2 independent experiments. (F) Suggested model of [Ca$^{2+}$]$_c$ modulation. Time frame I: Under steady-state conditions, Ca$^{2+}$ is released into the cytoplasm and pumped back into the ER. CD38$^+$ HSCs release more Ca$^{2+}$ from ER than CD38$^-$ cells due to the cADPR, the product of CD38 enzymatic activity. Time frame II: Blocking Ca$^{2+}$-ATPase pumping Ca$^{2+}$ into the ER with TG led to a faster rise in [Ca$^{2+}$]$_c$ concentration CD38$^+$ HSCs than in CD38$^-$ counterparts. Time frame III: Excessive [Ca$^{2+}$]$_c$ is removed from cytoplasm. (G) [Ca$^{2+}$]$_c$ dynamics in CD38$^-$ and CD38$^+$ HSCs. (H) Time of the peak [Ca$^{2+}$]$_c$ in CD38$^-$ and CD38$^+$ HSCs after addition of TG, $n = 3$. (I) Time of the peak [Ca$^{2+}$]$_c$ in CD38$^-$ and CD38$^+$ HSCs after addition of TG in the presence of a CD38 inhibitor, $n = 3$. $P$-values were calculated using the paired $t$ test; for panel E, the unpaired $t$ test was used. $^*p < 0.05$, $^{**}p < 0.01$. The data underlying this figure can be found in S1 Data. Graphical schemes in panels A and F were created with BioRender.com. cADPR, cyclic adenosine diphosphate ribose; ER, endoplasmic reticulum; LT-HSC, long-term hematopoietic stem cell.

8-Bromo-cADPR antagonist activated hHSC cell cycle entrance in coculture with CD38$^+$ progenitors (S6E Fig). To confirm that CD38 truly regulates [Ca$^{2+}$]$_c$ in HSCs, we used thapsigargin (TG), which inhibits Ca$^{2+}$ transport from the cytoplasm into the ER. As expected (Fig 6F), [Ca$^{2+}$]$_c$ increased significantly faster in CD38$^+$ HSCs compared to CD38$^-$ cells (Fig 6G and 6H) and this difference was abrogated by treatment with 78c, the CD38 inhibitor (Fig 6I), suggesting that Ca$^{2+}$ release from the ER in CD38$^+$ HSCs is mediated by CD38. Interestingly, we

did not observe a significant difference in calcium accumulation time between CD38+ and CD38KO HSCs (S5E Fig). This could be due to compensatory mechanisms at the organismal level in CD38KO mice. Previous study has reported that CD38KO mice are protected against glucose intolerance induced by a high-fat diet and have increased energy expenditure in homeostatic conditions, suggesting large differences in metabolism between wt and CD38KO mice [34]. Together, these data suggest that CD38-dependent cADPR but not ADPR production contributes to high $[Ca^{2+}]_c$ concentration in CD38+ HSCs, which maintains their quiescence.

## c-Fos regulates quiescence of CD38+ LT-HSCs

To clarify how the CD38/cADPR/$Ca^{2+}$ axis regulates HSC dormancy, we performed a bulk transcriptome RNA sequencing of CD38+ and CD38- LT-HSCs (LSK CD48- CD150+ CD34- CD201+) and found that while 205 genes were significantly down-regulated in CD38+ LT-HSCs, 225 were up-regulated (Fig 7A and S4 Table). Gene set enrichment analysis (GSEA) revealed a significant down-regulation of genes related to hematopoietic stem cell differentiation programs, mitochondrial respiratory chain complex assembly, and NADH dehydrogenase

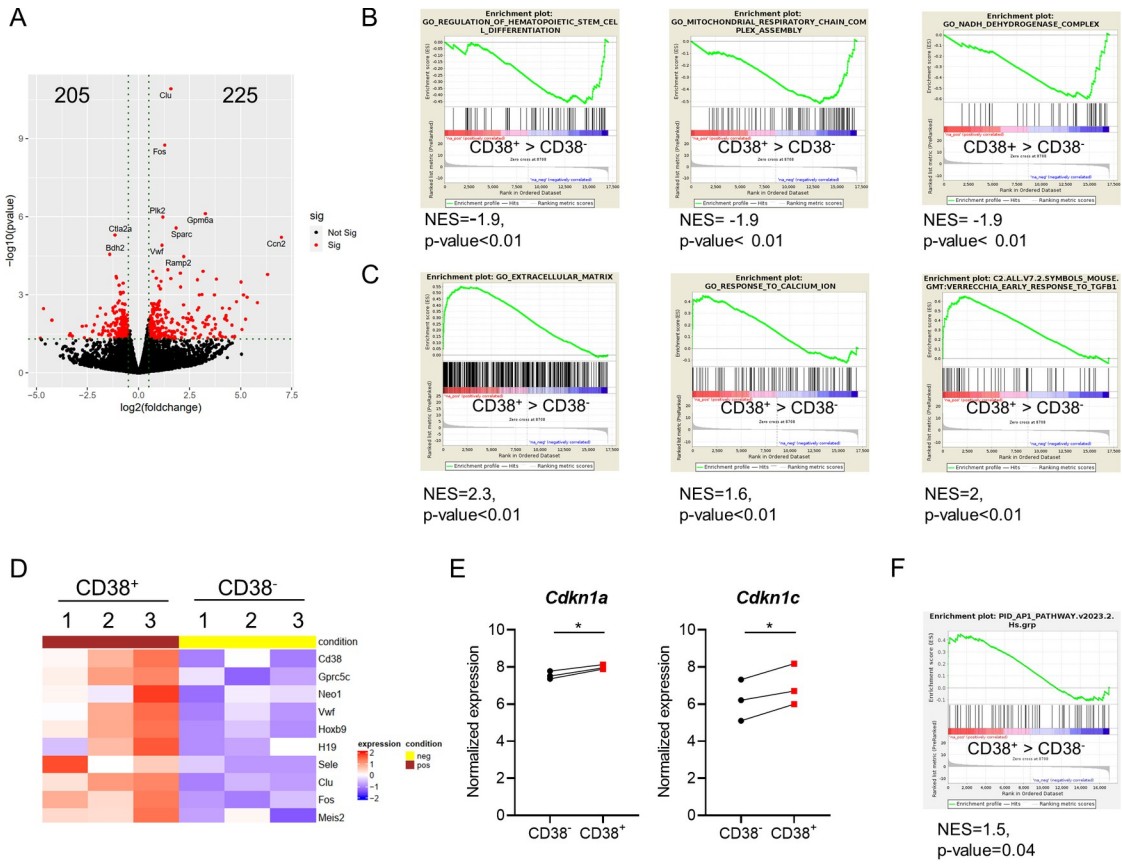

**Fig 7. c-Fos is up-regulated in CD38+ LT-HSCs.** (A) Volcano plot of differentially expressed genes in CD38+ LT-HSCs compared to CD38- cells. (B) GSEA of down-regulated genes in CD38+ LT-HSCs compared to CD38- stem cells. (C) GSEA of up-regulated genes in CD38+ LT-HSCs compared to CD38- cells. (D) Heatmap depicting dHSCs and cell cycle-related genes expressed in CD38+ and CD38- LT-HSCs. (E) Normalized expression of *Cdkn1a* and *Cdkn1c* in CD38+ vs. CD38- LT-HSCs. *P*-values were calculated using the paired *t* test, *$p < 0.05$. (F) GSEA of AP-1 pathway-related genes in CD38+ LT-HSCs compared to CD38- cells. The data underlying this figure can be found in S4 Table and S1 Data. GSEA, gene set enrichment analysis; LT-HSC, long-term hematopoietic stem cell.

complex in CD38$^+$ LT-HSCs compared to CD38$^-$ cells. Similarly, gene sets controlling the response to calcium ions, extracellular matrix interaction, and TGF-β1 response were up-regulated in CD38$^+$ LT-HSCs (Fig 7B and 7C). HSC-related genes, such as *Hoxb9*, *H19*, *VwF*, *Clu*, and *Sele* [14,35–38], as well as genes associated with HSC dormancy, namely, *Gprc5c*, *Meis2* [8], and *Neo1* [39], were up-regulated in CD38$^+$ LT-HSCs (Fig 7D). We did not find significant differences in *Cdk2*, *Cdk4*, *Cdk6*, and *CyclinD1* expression but CD38$^+$ LT-HSCs expressed the cell cycle inhibitors *Cdkn1a* and *Cdkn1c* at higher levels than CD38$^-$ LT-HSCs (Fig 7E). Intriguingly, the transcription factor *Fos*, whose expression was previously correlated to cell cycle activation [40], was one of the most significantly up-regulated genes in CD38$^+$ LT-HSCs. This observation was further corroborated by the fact that AP-1 complex responsive genes were enriched in CD38$^+$ LT-HSCs compared to CD38$^-$ counterparts (Fig 7F). Moreover, CD38$^+$ LT-HSCs displayed higher levels of transcriptionally active phosphorylated c-Fos (at Threonine 232, p-c-Fos) [41] than CD38$^-$ and CD38KO LT-HSCs (Figs 8A and S8A).

Correspondingly, single-cell tracking analysis revealed that blocking c-Fos interaction with DNA using a specific inhibitor, T-5224 [42], induced division of CD38$^+$ LT-HSCs but not CD38$^-$ cells (Fig 8B and 8C). Moreover, administering T-5224 to mice led to the partial loss of quiescence in CD38$^+$ HSCs but not in CD38$^-$ counterparts (Fig 8D). These data suggest that c-Fos transcriptional activity is necessary for CD38-mediated HSCs quiescence. As inhibiting the transcriptional activity of c-Fos affected cell cycle entrance of CD38$^+$ LT-HSCs but not CD38$^-$ cells, we hypothesized that CD38 regulates c-Fos expression via CD38/cADPR/Ca$^{2+}$ (Fig 8E). Indeed, treatment of HSCs with CD38 inhibitor or a cADPR antagonist (Br-cADPR) reduced the levels of active p-c-Fos (Figs 8E–8G and S8B), supporting the notion that the CD38/cADPR/Ca$^{2+}$ axis regulates c-Fos levels in CD38$^+$ LT-HSCs.

## CD38 controls p57$^{kip2}$ expression via c-Fos

To gain mechanistic insight into how CD38 and c-Fos regulate HSC dormancy, we analyzed the presence of c-Fos binding motifs in the regulatory regions of stem cell-related genes that were up-regulated in CD38$^+$ LT-HSCs (Fig 7D and 7E) and found that several genes, including *Cdkn1c*, a well-known regulator of HSC quiescence [19], have c-Fos binding motifs (S5 Table). Therefore, it is possible that c-Fos blocks cell cycle entrance of HSCs through *Cdkn1c* expression. Indeed, we confirm that not only expression of the *Cdkn1c* gene (Fig 7E) but also that of its gene product, p57$^{Kip2}$ protein, is higher in CD38$^+$ LT-HSCs than in CD38$^-$ cells (Fig 8H). Furthermore, treatment with cADPR antagonist or inhibiting c-Fos or CD38 enzymatic activity led to a reduction in p57$^{kip2}$ protein expression (Fig 8I–8K), thereby supporting our hypothesis that CD38 activates *Cdkn1c* expression in CD38$^+$ LT-HSCs via c-Fos.

## Discussion

In our study, the gene expression profile of single HSCs revealed that their heterogeneity is predominantly determined by genes related to the cell cycle, which is in agreement with previous findings [43]. Intriguingly, the cluster associated with quiescent state as well as with markers defining the most functional HSCs (Fgd5, Procr, and vWF) includes genes involved in pathways related to the activation of tumor necrosis factor alpha, up-regulation of interferon gamma and alpha response, Stat5 and Stat3 pathways. Our results are in agreement with the previous publication [8], which suggests that the intrinsic expression of interferon responsive genes in HSCs in a steady-state condition confers antiviral resistance to adult HSCs similar to what has been shown for human fetal HSCs and other stem cell types [44]. The up-regulation of TNFα signaling in this cluster is in agreement with other finding suggesting the pro-survival role of TNFα signaling in HSCs [45]. The up-regulation of Stat5 signaling in the cluster

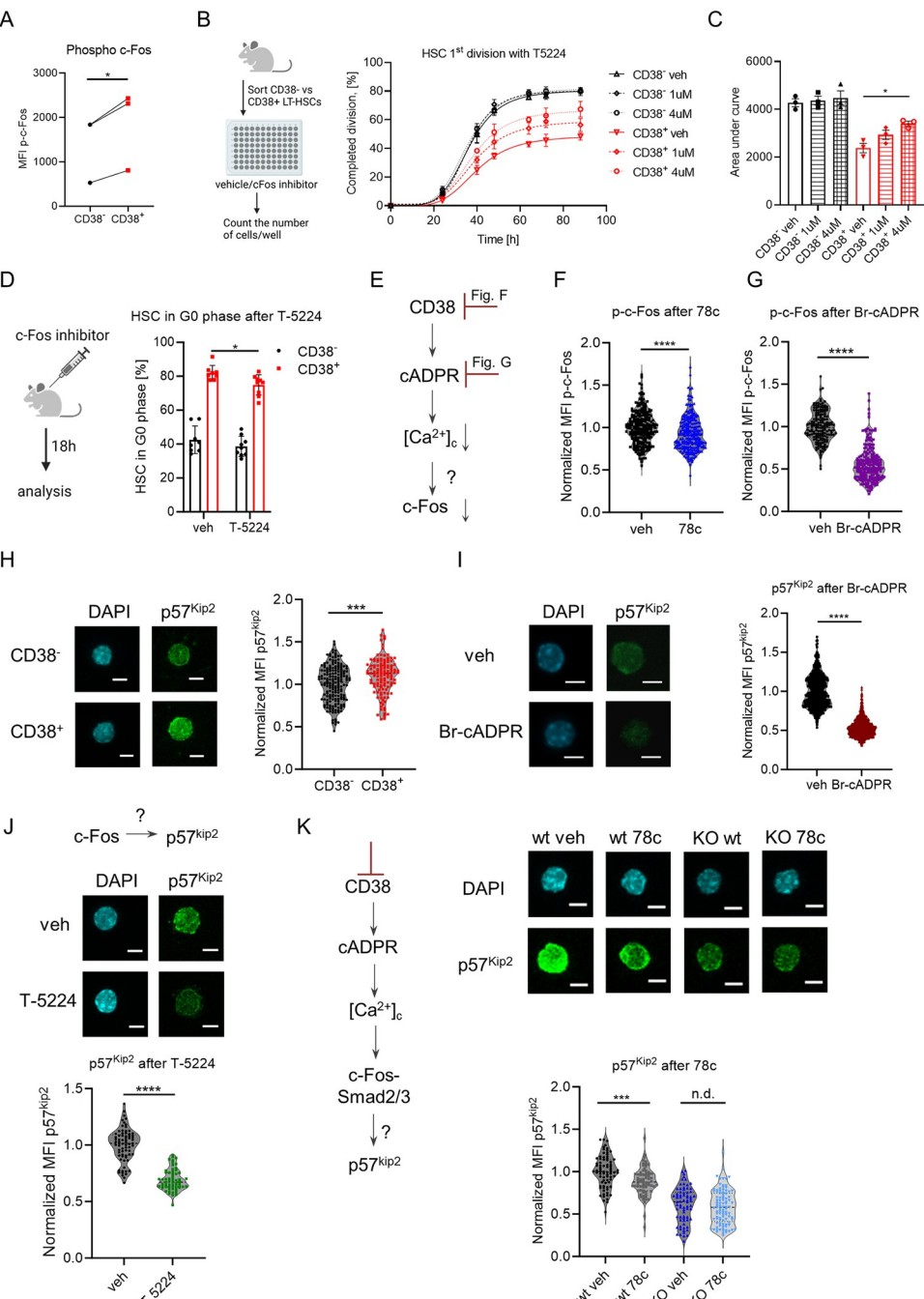

**Fig 8. c-Fos maintains CD38⁺ LT-HSC quiescence via p57^Kip2.** (A) MFI of intracellular p-c-Fos in CD38⁺ and CD38⁻ HSCs. *P*-value was calculated using the paired *t* test; *$p < 0.05$. (B) Single CD38⁻ and CD38⁺ LT-HSCs were sorted and cultured in liquid media with or without the c-Fos inhibitor T-5224. Frequency of LT-HSCs that had completed the first division during incubation is presented (3 independent experiments). (C) Quantification of AUC for data in panel B. *P*-value was calculated using the paired *t* test. (D) Cell cycle analysis of HSCs at 24 h after T-5224 injection (anti-Ki67 and DAPI staining was used), *n* = 8 vs. 9, 2 independent experiments. *P*-value was calculated using the Mann–Whitney *U*-test. (E) Set up of experiments for F and G. (F) p-c-Fos immunofluorescence in CD38⁺ LT-HSCs after 16 h with or without 78c in vitro (*n* = 247—veh, 223 - 78c, 2 independent experiments). (G) p-c-Fos immunofluorescence in HSCs after 24 h with or without Br-cADPR in vitro (*n* = 164—veh, 219—Br-cADPR, 2 independent experiments). (H) Representative maximum projection confocal images of total p57^kip2 and DAPI in individual CD38⁻ and CD38⁺ HSCs (left). Quantification of MFI of p57^kip2 immunofluorescence is shown (right) (*n* = 181—CD38⁻, 122—CD38⁺, 2 independent experiments). (I) Representative maximum projection confocal images of total p57^Kip2 and DAPI in individual CD38⁺ LT-HSCs after 16 h with or without Br-cADPR in vitro (left). Quantification of MFI of p57^kip2

immunofluorescence is shown (right) ($n$ = 696—veh, 1211—Br-cADPR). (J) Representative maximum projection confocal images of total p57$^{Kip2}$ and DAPI in individual HSCs after 24 h with or without c-Fos inhibitor in vitro (top). Quantification of MFI of p57$^{kip2}$ immunofluorescence is shown ($n$ = 72- veh, $n$ = 61—T-5224, 2 independent experiments). (K) Representative maximum projection confocal images of total p57$^{kip2}$ and DAPI in CD38$^{+}$ wt vs. CD38KO LT-HSCs after 24 h with or without 78c inhibitor in vitro (left). Quantification of MFI of p57$^{kip2}$ immunofluorescence is shown (right) ($n$ = 78/68/72/97 for wt veh/wt 78c/ko veh/ko 78c, respectively, 2 independent experiments). Scale bar is 5 μm. E–K—the $P$-values were calculated using unpaired $t$ test. *$p < 0.05$, **$p < 0.01$, ****$p < 0.0001$. The data underlying this figure can be found in S1 Data. Graphical schemes in panels B and D were created with BioRender.com. AUC, area under the curve; LT-HSC, long-term hematopoietic stem cell.

associated with the most functional HSCs is consistent with the previous publications showing the important role of Stat5 in HSC self-renewal [46] and quiescence support [47]. The importance of Stat3 signaling for HSC functionality (regulation of self-renewal, apoptosis, DNA repair, ROS, and internal IFN signaling) was shown in other recent work [48].

Although CD38 is associated with cell proliferation of different mature cells like T cells [49], neuronal cells [50], and fibroblasts [51], we found that *Cd38* expression in HSCs is inversely correlated with that of cell cycle activators but positively associated with the cell cycle inhibitor *Cdkn1c* and other well-known genes that define the most quiescent LT-HSCs, such as *VwF*, *Procr*, *Fgd5*, and *Gprc5c* [8,14,15,17]. In contrast to previous studies reporting that only CD38$^{+}$ HSPC compartment from adult mice contains LT-HSCs [52,53], we demonstrated using modern surface marker combinations for the isolation of LT-HSCs that while both populations: CD38$^{-}$ and CD38$^{+}$, can be classified as LT-HSCs, only CD38$^{+}$ LT-HSCs display characteristics of dormant HSCs [1]. Namely, our results indicate that CD38$^{+}$ LT-HSCs reside at the top of the hematopoietic hierarchy, that they have the highest repopulation capacity upon serial transplantation, and that they are the most quiescent stem cells under steady-state conditions. Hence, CD38 represents a marker that will help circumvent the limitations of the long-term label-retaining assays [1,7,26,54] or even negate the necessity of reporter mice [8,11] for studying the mechanisms underlying HSC dormancy.

Interestingly, low extracellular Ca$^{2+}$ has been demonstrated to support the ex vivo HSC maintenance [55]. We argue however, that such effect could potentially arise from the vastly different biology of HSCs cultured for an extended period (2 weeks) under normoxic conditions (approximately 20% O$_2$) in contrast to the physiology of HSCs within their native hypoxic BM niche (approximately 1% to 4% O$_2$ [56]). In the study by Luchsinger and colleagues [55], [Ca$^{2+}$]$_c^{low}$ HSCs displayed enhanced long-term repopulation capacity compared to [Ca$^{2+}$]$_c^{high}$ HSCs. However, the conclusion from this work appears contradictory when considering previous research from this group. In the earlier work, they suggested that HSC with higher CD150 protein levels are associated with higher [Ca$^{2+}$]$_c$ [57], whereas it was shown that CD150$^{hi}$ HSC have superior repopulation capacity compared to CD150$^{neg}$ HSCs [22]. However, when only dHSCs, and not bulk HSCs, were sorted based on their [Ca$^{2+}$]$_c$ levels into 2 groups for transplantation, an association between higher [Ca$^{2+}$]$_c$ and increased dHSC reconstitution potential was uncovered [11]. This agrees with our data that CD38$^{+}$ LT-HSCs are the most potent HSCs upon transplantation along with holding the higher [Ca$^{2+}$]$_c$ levels than CD38$^{-}$ LT-HSCs. Intriguingly, recent studies showed that elevating [Ca$^{2+}$]$_c$ of HSCs is important for triggering the cell cycle progression of HSCs under stress induced by 5-FU or LPS [11,58,59]. Therefore, high [Ca$^{2+}$]$_c$ may serve different roles across different cell cycle phases probably due to the different [Ca$^{2+}$]$_c$ threshold levels. Taken together, our results along with these findings suggest that while high [Ca$^{2+}$]$_c$ keeps HSCs dormant in the quiescent phase, higher [Ca$^{2+}$]$_c$ prompts HSCs to progress through the cell cycle phases. In conclusion, it is of great interest to the field to study how Ca$^{2+}$ regulates fate decisions of dHSCs in future research.

Moreover, we uncovered that CD38 cyclase activity is responsible for the high $[Ca^{2+}]_c$ in LT-HSCs and the up-regulation of *Fos* gene expression and phosphorylated c-Fos protein. Generally, *Fos* is known as one of the immediate early genes that is transiently expressed in stimulated cells, leads to cell cycle progression [60,61], and is a positive regulator of myeloid differentiation [62]. Besides, c-*Fos* is an oncogene, whose expression is often up-regulated in hematologic malignancies, e.g., in chronic and acute myeloid leukemia [63,64]. In our current work, we have shown for the first time that c-Fos regulates HSC dormancy under physiological conditions, uncovered how c-Fos is regulated by the enzymatic activity of CD38, and found that p57$^{Kip2}$ is a target for c-Fos. Therefore, c-Fos can activate multiple transcriptional programs in a cell type-specific manner, and its role in hematopoiesis regulation requires further investigation.

Additionally, we found that CD38 enzymatic activity on neighboring cells regulated the proliferation of CD38-negative hHSCs. In human BM several cell types express CD38: multipotent and restricted hematopoietic progenitors, plasma cells, activated T and B-lymphocytes, and NK cells [65]. Therefore, some of these CD38$^+$ cells can be the neighbors for hHSCs. Interestingly, several hematological malignancies (chronic myeloid leukemia, acute myeloid leukemia, acute lymphoblastic leukemia, and multiple myeloma) express CD38 at a high level [66]. We hypothesize that tumor microenvironment enriched in the products of CD38 ecto-enzymatic activity may keep healthy HSCs in the quiescent state leading to cancer-related pancytopenia [67] as well as it may preserve the dormancy of cancer stem cells leading to disease persistence. Therefore, a better understanding of the mechanisms controlling human HSC dormancy is required to support healthy hematopoiesis in patients with hematologic malignancies and develop more powerful strategies for cancer eradication.

In summary, we reveal that the CD38/cADPR/Ca$^{2+}$/c-Fos/p57$^{kip2}$ axis regulates HSC dormancy. Mechanistically, we demonstrate that CD38 itself regulates stem cell dormancy by shuttling intracellular $[Ca^{2+}]_c$ in a CD38/cADPR-dependent manner, which results in a consequent increase in c-Fos and the expression of the cell cycle inhibitor p57$^{kip2}$ (Fig 9). Manipulation of this axis can potentially stimulate dHSC expansion and their efficient response to hematopoietic stress.

## Material and methods

### Ethics statement

Animal experiments were performed in accordance with the German animal welfare legislation and were approved by the "Landesdirektion Sachsen" (TVV16/2017, TVV48/2022, TVV91/2017). The studies with human cells were approved by Ethikkommission an der TU Dresden (EK263122004, EK114042009).

### Reagents and resources

S6 Table lists all reagents used.

### Mice

C57BL/6N (B6), B6.SJL-Ptprc$^a$Pep3$^b$/BoyJ (SJL) were purchased from The Jackson Laboratory. C57BL/6N (B6) and B6.SJL-Ptprc$^a$Pep3$^b$/BoyJ (SJL) mice were crossed to produce F1 progeny (CD45.1/CD45.2) for transplantation experiments. To study the division history of HSCs, R26$^{rtTA}$/ *Col1A1*$^{H2B-GFP}$ mice were used [7]; the induction of H2B-GFP expression was performed as described in [26]. B6.129P2-Cd38$^{tm1Lnd}$/J (CD38KO) mice were obtained from Dr. Jaime Sancho and Dr. Frances Lund. Ki67$^{RFP}$ knock-in mice have been described previously

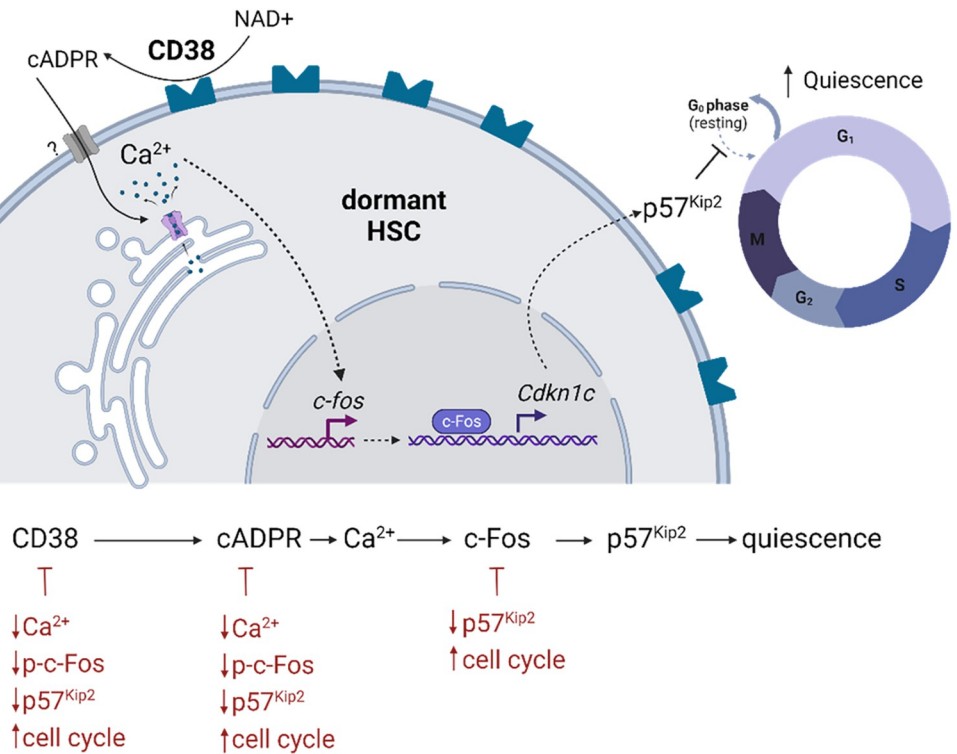

**Fig 9. Representative summary of findings.** Created with BioRender.com. cADPR, cyclic adenosine diphosphate ribose; HSC, hematopoietic stem cell; NAD, nicotinamide adenine dinucleotide.

[68,69]. All mice were bred and maintained under specific pathogen-free conditions in the animal facility at the Medical Theoretical Center of the Technical University, Dresden, Germany. Unless specified otherwise, 8- to 16-week-old mice of both genders were used for experiments.

## Cell isolation and flow cytometry

Cells were isolated from tibiae, femora, pelvis, and vertebrae by crushing bones in 5% fetal bovine serum (FBS) in phosphate-buffered saline (PBS) and passing them through a 40-μm filter. Erythrocytes were lysed using ACK Lysing Buffer. To calculate the amount of cells, cells isolated from 2 tibiae and 2 femora were stained with DAPI at 0.1 μg/ml and DAPI-negative cells were counted using MACSQuant Analyzer (Miltenyi Biotec). For sort, cells were stained with c-Kit bio antibody and Anti-Biotin MicroBeads were added to enrich for c-Kit$^+$ cells using LS columns. HSCs were defined as Lin$^-$ (negative for B220, CD3ε, CD19, NK1.1, Gr1, Ter119, and CD11b) Sca1$^+$ c-Kit$^+$ (LSK) CD48$^-$ CD150$^+$ cells. LT-HSCs: LSK CD48$^-$ CD150$^+$ CD34$^-$ CD201$^+$, MPP2: LSK CD48$^+$ CD150$^+$, MPP3/4: LSK CD48$^+$ CD150$^-$, and ST-HSCs: LSK CD48$^-$ CD150$^-$. Granulocyte-monocyte progenitors (GMPs) were defined as Lin$^-$ Sca1$^-$ c-Kit$^+$ (LK) CD16/32$^+$ CD150$^-$, pre-megakaryocyte progenitors, PreMeg: LK CD16/32$^-$ CD41$^+$ CD150$^+$, colony-forming unit-erythroid, CfuE: LK CD16/32$^-$ CD41$^-$ CD105$^+$ CD150$^-$, precolony-forming unit-erythroid, PreCfuE: LK CD16/32$^-$ CD41$^-$ CD105$^+$ CD150$^+$, megakaryocyte-erythroid progenitors, MEP: LK CD16/32$^-$ CD41$^-$ CD105$^-$ CD150$^+$, common myeloid progenitors, CMP: LK CD16/32$^-$ CD41$^-$ CD105$^-$ CD150$^-$. All analyses were performed on a BD LSR II, BD FACSAria II, BD LSRFortessa X-20, or a BD FACSCanto II (BD Bioscience). Data were analyzed using FlowJo software 10.7.1 (BD Bioscience).

## Single-cell RNA sequencing with 10x Genomics and analysis

LT-HSCs (LSK CD48$^-$ CD150$^+$, 3,000 cells) from 4 B6 mice were sorted into BSA-coated tube containing 5 μl of PBS with 0.04% BSA. All cells were carefully mixed with reverse transcription mix before loading them in a Chromium Single Cell A Chip on the 10x Genomics Chromium system [70] and processed further following the guidelines of the 10x Genomics user manual for Chromium Single Cell 3′ RNA-seq Library v2. In short, the droplets were directly subjected to reverse transcription, the emulsion was broken and cDNA was purified using silane beads. After amplification of cDNA with 12 cycles, it underwent a purification with 0.6 volume of SPRI select beads. After quality check and quantification, 20 μl cDNA was used to prepare single-cell RNA-seq libraries—involving fragmentation, dA-Tailing, adapter ligation, and a 13 cycles indexing PCR based on manufacturer's guidelines. After quantification, both libraries were sequenced on an Illumina Nextseq500 system in paired-end mode with 26 bp/57 bp (for read 1 and 2, respectively), thus generating approximately 50 to 80 mio. fragments for the transcriptome library on average. The raw count data generated from Cell Ranger pipeline was processed using Seurat v3.1 [12] by following the standard pipeline. Cells were filtered based on quality metrics (number of genes, total UMI counts, percentage of mitochondrial genes). Subsequently, filtered data were merged using "FindIntegrationAnchors" function of Seurat 3. For further analysis, merged data were log-normalized, regressed for library size, and percentage of mitochondrial genes and scaled. Cell cycle and dormancy scores were calculated with G2M and S phase genes from Seurat package and dormancy-related genes [8] (see S1 Table) and using "AddModuleScore" function of Seurat 3. For pseudotime trajectory analysis, standard pipeline of Monocle 2 [70] was used and dimensionality reduction was performed using "reduceDimension" function of Monocle 2 with following parameters: num_dim = 10, norm_method = "log", reduction_method = "tSNE". For unsupervised clustering of cells, we used clusterCells() function. To visualize gene modules showing similar kinetic trends, "plot_pseudotime_heatmap" function was used accounting a list of genes showing significant score for differential expression along pseudotime (q-value < 0.05) and genes that change as a function of pseudotime were grouped in 3 clusters.

## Gene ontology analysis

GO term analysis was performed using Enrichr [71]. Complete gene list per clusters were obtained by using the "crisp gene set." For visualization, statistically significant ($p$-value < 0.05) terms were selected from top 5 pathways (see S3 Table).

## LT-HSCs transplantation

For primary transplantation, 50 CD38$^-$ or CD38$^+$ LT-HSCs (LSK CD48$^-$ CD150$^+$ CD34$^-$ CD201$^+$) were sorted and transplanted together with $5 \times 10^5$ total BM competitor cells. For secondary transplantation, $5 \times 10^6$ CD45$^+$ total BM cells were transplanted into lethally irradiated recipients.

For LT-HSCs primary transplantation, 50 wt or CD38KO LT-HSCs (LSK CD48$^-$ CD150$^+$ CD34$^-$ CD201$^+$) were sorted and transplanted together with $5 \times 10^5$ total BM competitor cells. For secondary transplantation, donor HSCs (LSK CD48$^-$ CD150$^+$) were sorted and transplanted together with $5 \times 10^5$ total BM competitor cells into lethally irradiated recipients. For TBM primary transplantation, $1 \times 10^6$ CD45$^+$ total BM cells from wt or CD38KO mice were transplanted together with $1 \times 10^6$ total BM competitor cells. For secondary transplantation, $5 \times 10^6$ CD45$^+$ total BM cells were transplanted into lethally irradiated recipients.

For CD38KO transplantation, $7 \times 10^6$ CD45$^+$ total BM cells from CD38KO mice were transplanted into wt or CD38KO lethally irradiated recipients.

All recipient mice were lethally irradiated (9 Gy), and the cells were injected i.v.

## Cell cycle analyses

Cell cycle was analyzed using staining for Ki67 and DAPI as described before [72]. The groups of mice for each specific treatment (ctrl versus treatment at different time points) were analyzed on the same day, using the same amount of cells, the same master mix of antibody, and the same FACS machine and settings to compare ctrl versus treated mice. To label dHSCs, mice were injected with 1 mg of BrdU i.p. and kept with 0.8 mg BrdU per 1 ml in drinking water for 14 more days before sacrificing. Water was changed every 3 days. BrdU incorporation analysis was performed using anti-BrdU antibody as described before [21].

## In vivo drug administration

To study the effects of c-Fos inhibition, T-5224 in the vehicle (2% DMSO+30% PEG300+2% Tween80 in ddH2O) was injected i.p. at 30 mg/kg/mouse 18 h prior to analysis. Control mice were injected with the corresponding amount of vehicle. To mimic viral infection, pIC was administered i.p. at a dose of 5 mg/kg 18 or 48 h prior to analysis. To study HSC response to chemotherapy, 5-FU was injected i.p. at a dose of 150 mg/kg 4 or 8 days prior to analysis, and 78c was injected i.p. at a dose of 20 mg/kg/mouse, at days 3 to 7 after 5-FU.

## Blood counts

A Sysmex XT-3000 Vet automated hematology analyzer was used to measure blood cell counts.

## Intracellular calcium staining and flux

Calcium staining and flux were estimated by flow cytometry. Cells were incubated with 0.3 μm Fluo-8 AM for 1 h at room temperature or with 2 μm Indo-1 AM and 0.02% Pluronic F-127 in HBSS for 30 min at 37°C, washed, and resuspended in HBSS. Wavelength filters for 405 ± 20 nm (violet emission) and 530 ± 30 nm (450 LP filter, blue emission) were used to visualize $Ca^{2+}$-bound and -unbound dye ratio by flow cytometry, respectively. After recording baseline calcium, thapsigargin (TG, 1 mM) was added to the sample to induce $Ca^{2+}$ flux. Alternatively, 78c (1 μm) was added to cells 5 min before TG. The average ratio, R, of bound/unbound Indo-1 (405 nm/485 nm emission) was calculated.

## Mitochondrial membrane potential and mass

Cells were subjected to 250 nM tetramethylrhodamine ethyl ester (TMRE) to measure MMP or to 100 nM Mitotracker Green to measure mitochondrial mass in the presence of 50 μm Verapamil at 37°C for 20 min. Cells were washed and analyzed by flow cytometry.

## Single-cell LT-HSCs in vitro culture

Single long-term HSCs were sorted into 96-well plates containing StemSpan medium with 10 ng/ml SCF, 10 ng/ml THPO, 20 ng/ml IGF2, and 10 ng/ml FGF1, with or without 1 μm/4 μm c-Fos inhibitor T-5224, 0.1 μm CD38 inhibitor 78c, and cultured for 3 days at 37°C with 5% $CO_2$. The number of the cells per well was monitored twice daily under a light microscope.

## In vitro treatment of HSPCs

LSK cells from Ki67$^{RFP}$ mice were sorted and cultured ($5 \times 10^4$ per well) in StemSpan medium with 10 ng/ml SCF, 10 ng/ml THPO, 20 ng/ml IGF2, and 10 ng/ml FGF1 with or without 25

or 100 μm 8-Br-cADPR or 25–100 μm 8-Br-ADPR or 4 μm T-5224; 24 h later, cells were stained with anti CD48, CD150, Kit, Sca-1 antibodies, and 0.3 μm Fluo-8 AM. For experiments in Fig 8G and 8J, surface stained cells were fixed using eBioscience kit and HSCs were sorted on glass slides for immunofluorescent staining. For experiments in Fig 8F, 8I, and 8K, CD38+ LT-HSCs were sorted and cultured in StemSpan medium with 10 ng/ml SCF, 10 ng/ml THPO, 20 ng/ml IGF2, and 10 ng/ml FGF1 with or without 0.1 μm 78c or 25 μm 8-Br-cADPR; 16 or 24 h later, cells were fixed using eBioscience kit and put on glass slides for immunofluorescent staining.

## Immunofluorescent staining

HSCs sorted on glass slides were used for the immunofluorescent staining. Cells were blocked with 20% horse serum in 1X Permeabilization buffer (eBioscience) for 30 min at RT, stained with rabbit anti-phospho-c-Fos or rabbit anti-p57 antibodies for 2 h, washed and then incubated with secondary anti-rabbit AlexaFluor 488 antibody for 30 min. Cells were mounted using DAPI-containing mounting medium and sealed. Images were captured using Zeiss Observer Z.1 (ApoTome II) (ZEISS) or Leica TCS SP5 confocal microscope (Leica Microsystems) using 40× or 63× objective. From 6 to 8 z-stacks were taken per image and fluorescence was analyzed using Fiji [73].

## Human HSC in vitro culture

Bone marrow stem cell apheresates were collected from healthy donors at the Dresden Bone Marrow Transplantation Centre of the University Hospital Carl Gustav Carus. The donors fulfilled the standards for bone marrow donation (e.g., free of HIV, HBV, and serious illness), were informed and gave their approval. MOLM-13 were obtained from ATCC. BM aspirates were layered on top of Ficoll-Paque PLUS media and centrifuged at 800g for 20 min at 20°C (brake off), mononuclear cells (MNCs) were then isolated from buffy coat fraction in the interphase of Ficoll gradient. MNCs were stained for surface hHSCs markers: Lineage (CD3, CD14, CD16, CD19, CD20, CD56), CD34, and CD38. HSCs (Lin- CD38$^{lo/-}$ CD34$^+$) were sorted into 96-well plates in to CellGenix TM GMP SCGM medium supplemented with 2.5% FBS, human FLT3L, human SCF, and human IL-3 (all 2.5 ng/ml), and cultivated at 37°C 5% CO$_2$; 50,000 cells were seeded into the above-mentioned media with the same conditions in triplicates with or without 4 μm 78c or 100 μm Br-cADPR. Next day, the cells were stained with anti CD34, CD38 antibodies, fixed, permeabilized with eBioscience kit, and stained with anti-Ki67 antibody together with 5 μg/ml DAPI. Alternatively, MNCs were incubated with anti-CD34 MicroBeads and enriched for CD34+ cells using LS columns, and 3,000 HSCs (CD38$^{lo/-}$ CD34$^+$) were sorted into 96-well plates into CellGenix TM GMP SCGM medium supplemented with 2.5% FBS, human FLT3L, human SCF, and human IL-3 (all 2.5 ng/ml) containing $1 \times 10^5$ MOLM-13 cells, and cultivated at 37°C 5% CO$_2$ in duplicates with or without 1 μm 78c. Next day, the cells were stained with anti CD34, CD38 antibodies. Cell cycle was analyzed using staining for Ki67 and DAPI.

## Immunostaining of human bone marrow

Immunohistochemical staining was performed as described earlier on the Ventana Ultra Instrument (Ventana Medical Systems, Arizona, United States of America) [74]. Acquisition of the multispectral images was performed with the Vectra 3 automated imaging system (Akoya Biosciences). Data analysis was done in Imaris 9 software.

## Immunofluorescent staining of human cells

Lin[-] CD34[+] CD38[-]/Lin[-] CD34[+] CD38[+]/Lin[-] CD34[-] CD38[++] cells after fixation with eBioscience kit were sorted on glass slides and used for the immunofluorescent staining. Cells were blocked with Duolink Blocking solution (Sigma) for 30 min at RT, stained with anti-hCD38 APC in Duolink Antibody Diluent (Sigma) for 3 h, washed 3× with 1X Permeabilization buffer (BD). Cells were mounted using DAPI-containing mounting medium and sealed. Images were captured using Zeiss Observer Z.1 (ApoTome II) (ZEISS) using 63× objective. From 5 to 7 z-stacks were taken per image and fluorescence was analyzed using Fiji [73].

## CD38 cyclase activity

CD38 cyclase activity was measured according to [75] with minor changes. MOLM-13 cells were sonicated with 40W 20 kHz 3 times for 5 s on ice in a lysis buffer containing 10 mM Tris/HCl (pH 7.4), 0.25 M sucrose, 20 mM NaF, 1 mM DTT, 5 mM EDTA, 1 mM PMSF (all chemicals from Sigma–Aldrich), and a protease inhibitor cocktail (chymostatin, leupeptin, antipain, and pepstatin; Santa Cruz Biotechnology, sc-24948A). Fluorescence (ex = 300 nm, em = 410 nm) of reaction mix containing $1 \times 10^6$ lysed MOLM-13, 200 μm nicotinamide guanine dinucleotide sodium salt (NGD) and 0 to 1.28 μm 78c was recorded for 1 h. To generate dose-response curve, normalized relative fluorescence intensity at 1 h was plotted against log 78c concentration.

## Single-cell hLT-HSCs in vitro culture

Human MNCs were stained for surface LT-HSCs markers: Lineage (CD3, CD14, CD16, CD19, CD20, CD56), CD34, CD38, CD90, CD45RA. and single human LT-HSCs (Lin[-] CD38[lo/-] CD34[+] CD90[+] CD45RA[-]) were sorted into 96-well plates in to CellGenixGMP SCGM medium supplemented with 2.5% FBS, human FLT3L, human SCF, and human IL-3 (all 2.5 ng/ml) with or without 8 μm T-5224 or 4 μm 78c, and cultivated at 37˚C 5% $CO_2$. The number of the cells per well was monitored daily using a light microscope.

## Bulk RNA sequencing

A total of 2,000 LT-HSCs (LSK CD48[-] CD150[+] CD34[-] CD201[+]) that were CD38[+] or CD38[-] were sorted (pooled cells from 10 mice per sample). Bulk RNA sequencing was performed as previously described [72]. Illumina sequencing was done on a Nextseq500 with an average sample sequencing depth of 60 million reads.

## Transcriptome mapping

Low-quality nucleotides were removed with Illumina fastq filter (http://cancan.cshl.edu/labmembers/gordon/fastq_illumina_filter/). Reads were further subjected adaptor trimming using cutadapt [76]. Alignment of the reads to the mouse genome was done using STAR Aligner [77] using the parameters: "—runMode alignReads—outSAMstrandField intronMotif —outSAMtype BAM SortedByCoordinate—readFilesCommand zcat." Mouse Genome version GRCm38 (release M12 GENCODE) was used for the alignment. HTSeq-0.6.1p1 [78] was used to count the reads that map to the genes in the aligned sample files. Read Quantification was performed using the parameters: "htseq-count -f bam -s reverse -m union -a 20." The GTF file (gencode.vM12.annotation.gtf) used for read quantification was downloaded from Gencode [79].

### Differential expression analysis

Gene centric differential expression analysis was performed using DESeq2_1.8.1 [80]. Volcano plot was created using ggplot2_1.0.1 [81]. Heatmaps were generated using ComplexHeatmap package of R/Bioconductor [82].

### Gene enrichment analysis

Pathway and functional analyses were performed using GSEA [83]. GSEA is a stand-alone software with a GUI. To run GSEA, a ranked list of all the genes from DESeq2-based calculations was created by taking the -log10 of the *p*-value and multiplying it with the sign the of the fold change. This ranked list was then queried against MsigDB [84].

### Transcription factor binding site prediction

"runFimo" command of "memes" https://bioconductor.org/packages/memes package (from R) was run of the 3Kb upstream regions of the genes using the JASPAR2020 database [85].

### Data

The authors confirm that all relevant data underlying the findings are within the paper and its supporting files. Bulk and single-cell RNA-sequencing data are available at GEO under accession numbers GSE196760 and GSE196759, respectively. All data related to the figures can be found in S1 Data.

### Statistics

Data are presented as mean ± SEM. Significance was calculated using the Mann–Whitney *U*-test, unless stated otherwise. All statistical analyses were performed using GraphPad Prism 8.2.1 for Windows (GraphPad Software, La Jolla, California, USA; www.graphpad.com).

### Supporting information

**S1 Fig. Single-cell transcriptome analysis of HSCs.** (A) Uniform manifold approximation projection (UMAP) representation depicting the transcriptional profiles of individual HSCs (LSK CD48⁻ CD150⁺), clustering base on cell cycle-related genes. (B) Aligned kinetic curves for selected cell cycle-related genes along pseudotime. (C) Aligned kinetic curves for selected HSCs' dormancy-related genes along pseudotime. The data underlying this figure can be found in S1 Table.
(TIF)

**S2 Fig. Characterization of CD38⁺ and CD38⁻ HSPCs.** (A). Gating strategy for analysis of murine HSPCs. (B) Flow cytometry analysis of CD38 expression in HSPCs compartment (MPP3/4: Lin⁻ Sca-1⁺ Kit⁺ (LSK) CD48⁺ CD150⁻, MPP2: LSK CD48⁺ CD150⁺, ST-HSCs: LSK CD48⁻ CD150⁻, HSCs: LSK CD48⁻ CD150⁺). HSPCs from CD38KO were used as negative control for gating. (C) Gating strategy for defining CD38⁺ fraction using total BM cells. (D) FACS analysis of defined markers surface expression on CD38⁻ and CD38⁺ HSCs.
(TIF)

**S3 Fig. Comparison of CD38⁺ and CD38⁻ LT-HSCs.** (A). Reanalysis of sorted LT-HSCs used for transplantation experiments. (B) Frequency of T, B and myeloid cells in donor-derived WBC at the different time points after primary and secondary transplantation (Fig 2D–2H). (C and D) Surface expression of CD38 in donor-derived HSCs at 20 weeks after primary transplantation of CD38⁺ or CD38⁻ LT-HSCs, data from individual mice containing >60 cells in

the donor HSC population in the recorded fcs file, 2 independent experiments. (E) HSC mitochondrial membrane potential analysis ($n = 5$). (F) Analysis of mitochondrial mass in HSCs using MitoTracker Green (MTG) in the presence of verapamil ($n = 4$). (G) Representative FACS plots for discrimination of G0, G1, and SG2M phases of the cell cycle of LT-HSCs. $P$-values were calculated using the paired $t$ test, *** $p < 0.001$. The data underlying this figure can be found in S1 Data.
(TIF)

**S4 Fig. Analysis of CD38KO mice.** (A) Gating strategy for analysis of restricted myeloid progenitors. (B) Number of RBC in peripheral blood of wt and CD38KO (KO) mice. (C) Number of platelets (PLT) in peripheral blood of wt and CD38KO mice. (D) Number of WBC in PB of wt and CD38KO mice. (E) Number of restricted progenitors in bone marrow of wt and CD38KO mice. (F) Number of HSPCs in bone marrow of wt and CD38KO. For panels B–F, $n = 6$ wt, 8 KO. (G) Experimental setup for transplantation of TBM from CD38KO mice to wt and CD38KO. Created with BioRender.com. Cell cycle analysis of donor HSCs 8 weeks after transplantation using Ki67 and DAPI staining ($n = 5$ wt, 4 KO). $P$-values were calculated using Mann–Whitney test. The data underlying this figure can be found in S1 Data.
(TIF)

**S5 Fig. Role of CD38 in activation of LT-HSC proliferation.** (A) Setup for single-cell division tracing experiment. Single CD38$^+$ LT-HSCs from wt and LT-HSCs from CD38KO mice were sorted and cultured in liquid media with or without 78c. Created with BioRender.com. (B) Frequency of LT-HSCs that had completed the first division during incubation time is presented (4 independent experiments). (C) Quantification of AUC for B. (D) CD38$^+$ HSCs from Ki67-RFP reporter mice were sorted and incubated with or without 78c ($n = 3$). Frequencies of RFP$^+$ cells were analyzed 24 h later. (E) Calcium flux assay for wt and CD38KO HSCs. Time of the peak in HSC $[Ca^{2+}]_c$ after addition of TG, $n = 4$ vs. 5. $P$-values were calculated using paired $t$ test, *$p < 0.05$. The data underlying this figure can be found in S1 Data.
(TIF)

**S6 Fig. CD38 regulates hHSC division.** (A) Gating strategy for isolation of human HSCs and LT-HSCs. (B) Representative immunofluorescence intracellular staining of human cells and quantification of CD38 integrated fluorescence intensity ($n = 43/63/31$ for CD38$^-$ CD34$^+$, CD38$^+$ CD34$^+$, CD38$^{++}$ CD34$^-$, respectively). (C) Surface expression of CD38 on MOLM-13. Left–isotype ctrl, right–anti-CD38 antibody. (D) Dose-response curve for cyclase activity of MOLM-13 lysates in response to CD38 inhibitor ($n = 3$). Sigmoidal standard curve was interpolated, 95% confidence interval is shown. (E) Lin$^-$ CD34$^+$ human cells were sorted and incubated 24 h with or without Br-cADPR and 78c, cell cycle of CD38$^{lo/-}$ CD34$^+$ cells was analyzed using Ki67 staining. Graphical scheme was created with BioRender.com. $P$-values were calculated using the unpaired $t$ test, *$p < 0.05$. The data underlying this figure can be found in S1 Data.
(TIF)

**S7 Fig. ADPR does not influence the cell cycle of HSCs.** (A) LSK from Ki67$^{RFP}$ reporter mice were sorted and cultured for 24 h in the presence of ADPR antagonist (8-Br-ADPR). Relative $[Ca^{2+}]_c$ concentration in HSCs treated with 0–100 μm 8-Br-ADPR, ($n = 4$). (B) Frequency of Ki67-RFP$^+$ HSCs 24 h after treatment with 0–100 μm Br-cADPR, ($n = 4$). Multiple-group comparisons were performed using Brown–Forsythe and Welch ANOVA followed by Dunnett's T3 multiple comparison tests. (C) Representative histogram of intracellular p-c-Fos in CD38$^-$ and CD38$^+$ HSCs. The data underlying this figure can be found in S1 Data.
(TIF)

**S8 Fig. Inhibition of c-Fos activity in mouse and human HSCs.** (A) Quantification of p-c-Fos MFI in LT-HSC from wt and LT-HSCs from CD38KO ($n = 247$—wt, $n = 268$—CD38KO). (B) Quantification of p-c-Fos MFI in CD38KO LT-HSC cultured for 24 h with or without 78c ($n = 40$—wt, $n = 54$—CD38KO). A, B—the $P$-values were calculated using unpaired $t$ test, ****$p < 0.0001$. (C) Set-up of single-cell tracing experiment. Human single LT-HSCs were sorted into plate and incubated in the presence of c-Fos (T-5224) or CD38 (78c) inhibitors. Numbers of cells in wells were detected. Created with BioRender.com. (D) Frequency of LT-HSCs that had completed the first division during incubation time is presented for 1 donor. (E) Quantification of the area under the curve (AUC) for data in panel D for 3 donors. Multiple-group comparison was performed using Brown–Forsythe and Welch ANOVA followed by Dunnett's T3 multiple comparison tests. The data underlying this figure can be found in S1 Data.
(TIF)

**S1 Table. Cell cycle and dormancy genes.**
(XLSX)

**S2 Table. Genes in clusters along pseudotime.**
(XLSX)

**S3 Table. Pathway analysis for clusters of genes along pseudotime.**
(XLSX)

**S4 Table. Differently expressed genes in CD38+ vs. CD38- LT-HSCs.**
(XLS)

**S5 Table. c-Fos binding motifs in up-regulated stem cell-related genes in CD38+ LT-HSCs.**
(XLSX)

**S6 Table. Reagents and resources.**
(XLSX)

**S1 Data. Numerical values for all main and supporting figures.**
(XLSX)

## Acknowledgments

Graphical schemes were created using BioRender.com. We thank Anja Krüger and Robert Kuhnert for technical assistance, Core Facility Cellular Imaging at Faculty of Medicine, TU Dresden as well as Deep Sequencing Facility, DRESDEN-*concept* Genome Center. English language and content editing was provided by Vasuprada Iyengar.

## Author Contributions

**Conceptualization:** Ben Wielockx, Tatyana Grinenko.

**Formal analysis:** Liliia Ibneeva, Sumeet Pal Singh, Anupam Sinha, Sema Elif Eski, Rebekka Wehner, Luise Rupp, Iryna Kovtun, Alexander Gerbaulet, Tatyana Grinenko.

**Funding acquisition:** Ben Wielockx, Tatyana Grinenko.

**Investigation:** Liliia Ibneeva, Sumeet Pal Singh, Rebekka Wehner, Luise Rupp, Susanne Reinhardt, Andreas Dahl, Tatyana Grinenko.

**Methodology:** Juan Alberto Pérez-Valencia, Susanne Reinhardt, Manja Wobus, Malte von Bonin, Andreas Dahl, Marc Schmitz, Tatyana Grinenko.

**Project administration:** Martin Bornhäuser, Triantafyllos Chavakis, Ben Wielockx, Tatyana Grinenko.

**Resources:** Alexander Gerbaulet, Manja Wobus, Malte von Bonin, Jaime Sancho, Frances Lund, Marc Schmitz, Martin Bornhäuser, Tatyana Grinenko.

**Supervision:** Marc Schmitz, Martin Bornhäuser, Tatyana Grinenko.

**Validation:** Sumeet Pal Singh, Anupam Sinha, Andreas Dahl, Tatyana Grinenko.

**Visualization:** Liliia Ibneeva, Anupam Sinha, Sema Elif Eski, Iryna Kovtun, Tatyana Grinenko.

**Writing – original draft:** Liliia Ibneeva, Triantafyllos Chavakis, Ben Wielockx, Tatyana Grinenko.

**Writing – review & editing:** Liliia Ibneeva, Alexander Gerbaulet, Triantafyllos Chavakis, Ben Wielockx, Tatyana Grinenko.

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
