## [Editor Report · Decision Letter 0]

19 Jul 2023

Dear Dr Grinenko, 

Thank you for submitting your manuscript entitled "CD38 promotes hematopoietic stem cell dormancy" for consideration as a Research Article by PLOS Biology.

Your manuscript has now been evaluated by the PLOS Biology editorial staff as well as by an academic editor with relevant expertise and I am writing to let you know that we would like to send your submission out for external peer review.

Once your full submission is complete, your paper will undergo a series of checks in preparation for peer review. After your manuscript has passed the checks it will be sent out for review. To provide the metadata for your submission, please Login to Editorial Manager (https://www.editorialmanager.com/pbiology) within two working days, i.e. by Jul 21 2023 11:59PM.

Kind regards,

Luke

Lucas Smith, Ph.D.

Senior Editor

PLOS Biology

lsmith@plos.org

---

## [Decision Letter · Decision Letter 1]

3 Oct 2023

Dear Dr Grinenko,

Thank you for your patience while your manuscript "CD38 promotes hematopoietic stem cell dormancy" was peer-reviewed at PLOS Biology. It has now been evaluated by the PLOS Biology editors, an Academic Editor with relevant expertise, and by several independent reviewers. In light of the reviews, which you will find at the end of this email, we would like to invite you to revise the work to thoroughly address the reviewers' reports.

As you will see below, the reviewers find the study interesting, but they have identified a number of issues and provide suggestions to strengthen the paper and bring the conclusions more closely in line with the data. We think these will need to be addressed before we can consider your manuscript further for publication. As a note, our Academic Editor has commented that some of Reviewer 2's requests could be addressed by analyses of human RNA seq datasets to corroborate the statement that human HSCs at all stages of development are CD38- -- but noting this, simple extrapolation from mouse to human is likely inappropriate and this should be clarified. 

Given the extent of revision needed, we cannot make a decision about publication until we have seen the revised manuscript and your response to the reviewers' comments. Your revised manuscript is likely to be sent for further evaluation by all or a subset of the reviewers.

**IMPORTANT - SUBMITTING YOUR REVISION**

*Re-submission Checklist*

*Published Peer Review*

*PLOS Data Policy*

*Blot and Gel Data Policy*

Sincerely,

Lucas

Lucas Smith, Ph.D.

Senior Editor

PLOS Biology

lsmith@plos.org

REVIEWS:

Reviewer #1: In this study, Ibneeva et al demonstrate a role for CD38 both in defining the dormant subset of murine LT-HSC, and functionally in maintenance of HSC dormancy through an axis that involves CD38 enzymatic activation of cADPR, Ca2+ flux, and downstream cFos-mediated expression of the cell cycle regulator, p57Kip2. Overall, the experiments provide strong evidence for this mechanism by leveraging multiple complementary techniques and robust serial transplantation assays. The manuscript is well-written, easy to follow, and concise. The findings are of significant impact by providing novel mechanistic insight into the regulation of HSC dormancy, linking complementary findings from previously published manuscripts, as well as identification of a novel surface marker for the most dormant subset of murine LT-HSC. Although some evidence is provided to suggest a similar (non-cell autonomous) role for CD38 in human HSCs, the precise role in human HSC dormancy remains to be shown conclusively through functional analysis. However, given the breadth of the current study, I do not feel additional human HSC experiments are necessarily in scope, but would be intriguing to study further in future studies. Overall, I believe the manuscript warrants publication in PLOS Bio as it may be of broad interest to the hematopoiesis and stem cell research communities.

I have only several minor comments that should be addressed prior to publication:

1) Line 85: Sorting/gating strategy for LT-HSCs used for scRNAseq should be included in the data in Fig 1 or as a supplemental figure. If obtained, post-sort purity checks should also be shown.

2) Figure 1 should include how cells clustered in UMAP space, or this should be included in the supplement, for reference.

3) Figure 1A: How was pseudotime ordering determined? The root of pseudotime is often chosen arbitrarily. If that is the case, it might be more intuitive to choose the dormant HSC state as the root.

4) Fig 2D: Post-sort purity checks by FACS, if performed, should be shown for all sorting experiments to assess purity of sorted populations for transplant.

5) Line 220: Please explain what is the putative neighboring CD38+ cells in the context of hHSC in the marrow niche in vivo. Are these proposed to be stromal populations, or more differentiated hematopoietic progenitors?

6) Figure 4I: These experiments should also be performed in the absence of CD38+ MOLM13 cells, to show that the 78c inhibitor has no effect on cell cycle in the absence of paracrine CD38-expressing cells. Also were MOLM13 lysates sufficient to affect cell cycle in these assays (since Fig S4 shows CD38 activity presence in the MOLM13 lysates that could be inhibited by 78c)?

7) Figure 3: Human LT-HSCs are more highly purified within the CD34+CD38-/lo population based on expression of additional markers (such as CD90, CD49f, CD201). The authors should comment on the limitations of the analysis in figure 3 in that the CD34+38-/lo population is likely to be heterogeneous in regards to dormancy.

8) Figure 4G: Did [Ca2+]c also increase significantly faster in CD38 wt compared with CD38 KO LT-HSCs, as seen for CD38+ vs CD38- cells?

9) Figure 6: Did Br-cADPR treatment also decrease p57kip2 MFI?

Reviewer #2, Tsuyoshi Fukushima (note, Reviewer 2 has signed this review): This is an interesting paper showing that cytoplasmic calcium may be involved in HSC dormancy and the possibility that this regulation is mediated by CD38, cADPR, Fos, p57. Overall, the paper would be better if more is learned about the importance of CD38/cytoplasmic culcium to the physiological significance. 

major coment

① CD38 has a similar behavioral expression as other stem cell markers. Sorting for CD38+ will sort fractions with high levels of these stem cell markers.This means that the comparison of CD38+ and CD38- indicates that HSCs are enriched in the CD38+ fraction.

Therefore, it is necessary to verify whether CD38 is important for dormancy and stemness in the CD38 KO mice.

It would be important to analysis cell cycle, label retaining, evaluation of stress systems such as LPS or 5FU, and aging in the CD38 KO mice.

② To determine the extent how environmental CD38 is affecting HSCs, CD38KO HSC can be transplantated into CD38KO and WT recipients.

Also, if there is a difference in this experiment, To explain the degree of dependence of the environment and the HSCs themselves on CD38 by transplanting CD38 KO and WT HSCs to CD38 KO and WT rescipient.

③ In humans, can you explain the selectivity for HSCs, why HSCs specifically are dormant when cADPR is of environmental origin?

The importance of the production of cADPR by HSCs themselves in humans can be evaluated by analysis of cell cycle or division time by addition of 8-Br-cADPR.

If there is a difference in the above results, CD38 may not be on the surface in humans, but may be present intracellularly. It may be in the endoplasmic reticulum, nuclear membrane, or lysosomes, but cADPR could still be produced. It might be interesting to look at localization to know the possibilities. There may also be the possibility of other ADPR cyclases such as CD157 (BST1).

④ Is the Fos is activated in CD38+ also seen in RNAseq? For example, activation of AP1target, gene with AP1 motif, etc.

(Figure 5D is a gene with a cFos motif in a CD38+ gene with high expression. So you should check GSEA or GO etc for cFos motif gene etc.)

Also, does the 78c and 8-Br-cADPR treat decrease Fos target gene expression (including p57)?

minor coment

①It is easier to understand if the Psudo time scale is reversed.

② In vivo administration of BrdU for more than 1 day seems to induce cytotoxicity and stress responce. Differences in responsiveness to stress may be appreciated, but should be evaluated with caution.

③ Data from CD38 KO transplantation evaluated the importance of the CD38/calcium pathway in hematopoietic stem cells. This data is so important to this paper that it should be a main Figure.

Reviewer #3: Ibneeva et al., identify CD38 as a marker of dormant HSCs initially via transcriptomic profiling of single HSCs. They further show that the CD38 is enzymatically active on HSCs. The authors attempt to demonstrate that CD38 regulates cFos levels and p57kip2 protein via cFos in dormant HSCs. Although the data on CD38 is interesting, the data on CD38 regulation of cFos and p57 is greatly limited and preliminary. My specific comments are below: 

1) It is unclear and confusing how the initial analysis was performed; it should be clarified: "We observed that cells in the S (Fig. 1B) and the G2/M phases (Fig. 1C) were clustered together and that, as expected, most of the HSCs were quiescent (Fig. 1D)". Did the authors determine HSCs in S, G2M and G0 based on Seurat analysis? Did they compare the bioinformatic analysis to functional properties?

2) The authors state: "Notably, this cluster was characterized by genes involved in pathways related

to the activation of tumor necrosis factor alpha (TNFα) signaling, interferon gamma and alpha

response, Stat3 and Stat5, as well as transforming growth factor beta 1 (TGF-β1) signaling,

which is a well-known regulator of HCS quiescence". This cluster seems to be a collection of genes important for both HSC quiescence and activation. 

3) The data presented using available assays such as BrdU and analyses of H2BGFP suggest that CD38+ LT-HSCs are enriched for dormant HSCs but not that they are dormant HSCs. I would suggest instead of labeling "CD38+ LT-HSCs as dHSCs" , describing CD38+LT-HSCs as relatively enriched in dormant HSCs as compared to CD38- LT-HSCs. 

4) Experiments using 78c would benefit from using CD38 KO HSCs. 

5) The statement "and high cytoplasmic Ca2+ has been shown to support quiescence of HSCs" and the Fukushima et al., publication that the authors cite is debated as two other publications (PMID: 29946000; PMID: 31178255) that the authors did not cite, show that increased Calcium triggers HSCs activation. The authors should be cautious and discuss fully their results in the context of all previously published results and not a selected subgroup. 

6) It is intriguing that the authors do not see differences in CDK4/6 between CD38+ and CD38- HSCs; are there discrepancies between Fig 1G and Fig 5G? Have the authors considered that CD38+ cells may be an intermediate population between deep HSC dormancy to activation? 

7) The data on CD38 regulating cFos and p57 is preliminary, indirect and relatively weak. It might be better to move this data as supportive finding to the Supplement and modify the related statements. 

8) The title: "CD38 promotes HSC dormancy" is an overstatement; there is very limited data/if any in this work suggesting "promoting HSC dormancy". The data show that "CD38 enriches for dormant HSCs". 

9) The abstract should be modified accordingly. 

10) The authors should consider replacing the statement "more quiescent" by "more deeply in a state of quiescence".

---

## [Editor Report · Decision Letter 2]

12 Jan 2024

Dear Dr Grinenko,

Thank you for your patience while we considered your revised manuscript "CD38 promotes hematopoietic stem cell dormancy" for publication as a Research Article at PLOS Biology. This revised version of your manuscript has been evaluated by the PLOS Biology editors and the Academic Editor, who is satisfied by the response to reviewers and the changes made in the revision. 

Based on our Academic Editor's assessment of your revision, we are likely to accept this manuscript for publication. However, before we can editorially accept your study, we need you to to address the following data and other policy-related requests.

**IMPORTANT: Please address the following editorial requests: 

1) ETHICS STATEMENT: Thank you for providing an ethics statement in your methods section. Can you please update this to specify which local ethics commission approved the human cell studies?

2) METHODS: I noticed that some of the methods section has been included in the supplement. Please move all the methods into the main text.

3) CODE: Per journal policy, if any code was generated that is important to support the conclusions of your manuscript, we would require that you make it available without restrictions upon publication. Please ensure that any code is sufficiently well documented and reusable, and that your Data Statement in the Editorial Manager submission system accurately describes where your code can be found.

3) DATA: You may be aware of the PLOS Data Policy, which requires that all data be made available without restriction: http://journals.plos.org/plosbiology/s/data-availability. For more information, please also see this editorial: http://dx.doi.org/10.1371/journal.pbio.1001797

I see that you have provided the RNA-seq data as depositions to the GEO repository, which is great. However, to be fully compliant with our policy, we need you to also provide the all underlying data used to generate the other figures presented in the study. Note that we do not require all raw data. Rather, we ask that all individual quantitative observations that underlie the data summarized in the figures and results of your paper be made available in one of the following forms:

a. Supplementary files (e.g., excel). Please ensure that all data files are uploaded as 'Supporting Information' and are invariably referred to (in the manuscript, figure legends, and the Description field when uploading your files) using the following format verbatim: S1 Data, S2 Data, etc. Multiple panels of a single or even several figures can be included as multiple sheets in one excel file that is saved using exactly the following convention: S1_Data.xlsx (using an underscore).

b. Deposition in a publicly available repository. Please also provide the accession code or a reviewer link so that we may view your data before publication. 

>>Regardless of the method selected, please ensure that you provide the individual numerical values that underlie the summary data displayed in the following figure panels as they are essential for readers to assess your analysis and to reproduce it:

Fig 2B,C,E-H; Fig 3A-B,E,F-G; Fig4B-E,G-L; Fig 5B-F; Fig 6B-E,G-I; Fig 8A-K; Fig S3B-F; Fig S4B-G; Fig S5 B-E; Fig S6 B-E; Fig S7A-B; Fig S8A-B,D-E;

>>Please also ensure that figure legends in your manuscript include information on where the underlying data can be found, and ensure your supplemental data file/s has a legend.

>>Please ensure that your Data Statement in the submission system accurately describes where your data can be found.

We expect to receive your revised manuscript within two weeks. 

*Published Peer Review History*

*Press*

Sincerely,

Luke

Lucas Smith, Ph.D.

Senior Editor,

lsmith@plos.org,

PLOS Biology

---

## [Editor Report · Decision Letter 3]

24 Jan 2024

Dear Dr Grinenko,

Thank you for the submission of your revised Research Article "CD38 promotes hematopoietic stem cell dormancy" for publication in PLOS Biology and for addressing our last editorial requests in this revision. On behalf of my colleagues and the Academic Editor, Connie J Eaves, I am pleased to say that we can in principle accept your manuscript for publication, provided you address any remaining formatting and reporting issues. These will be detailed in an email you should receive within 2-3 business days from our colleagues in the journal operations team; no action is required from you until then. Please note that we will not be able to formally accept your manuscript and schedule it for publication until you have completed any requested changes.

**IMPORTANT: As a note - when assessing your revision, I saw that all of the requested changes were made in the 'track changes' version of the manuscript, but that the 'clean' version of the manuscript had not been updated. I have therefore swapped the old , 'clean' version of your manuscript with the updated, 'track changes' version. This means you will need to go through the manuscript and remove any highlighting, before publication. Please do also double check that everything else looks good with your submission after this change. 

PRESS

Sincerely, 

Lucas Smith, Ph.D.,

Senior Editor

PLOS Biology

lsmith@plos.org